# Morphometric analysis of lungfish endocasts elucidates early dipnoan palaeoneurological evolution

Alice M Clement[1]*, Tom J Challands[2], Richard Cloutier[3], Laurent Houle[3], Per E Ahlberg[4], Shaun P Collin[5], John A Long[1]

[1]College of Science and Engineering, Flinders University, Adelaide, Australia; [2]School of Geosciences, University of Edinburgh, Edinburgh, United Kingdom; [3]Département de Biologie, Chimie et Géographie, Université du Québec à Rimouski, Rimouski, Canada; [4]Subdepartment of Evolution and Development, Department of Organismal Biology, Uppsala University, Uppsala, Sweden; [5]School of Life Sciences, La Trobe University, Melbourne, Australia

*For correspondence:
alice.clement@flinders.edu.au

Competing interest: The authors declare that no competing interests exist.

**Abstract** The lobe-finned fish, lungfish (Dipnoi, Sarcoptergii), have persisted for ~400 million years from the Devonian Period to present day. The evolution of their dermal skull and dentition is relatively well understood, but this is not the case for the central nervous system. While the brain has poor preservation potential and is not currently known in any fossil lungfish, substantial indirect information about it and associated structures (e.g. labyrinths) can be obtained from the cranial endocast. However, before the recent development of X-ray tomography as a palaeontological tool, these endocasts could not be studied non-destructively, and few detailed studies were undertaken. Here, we describe and illustrate the endocasts of six Palaeozoic lungfish from tomographic scans. We combine these with six previously described digital lungfish endocasts (4 fossil and 2 recent taxa) into a 12-taxon dataset for multivariate morphometric analysis using 17 variables. We find that the olfactory region is more highly plastic than the hindbrain, and undergoes significant elongation in several taxa. Further, while the semicircular canals covary as an integrated module, the utriculus and sacculus vary independently of each other. Functional interpretation suggests that olfaction has remained a dominant sense throughout lungfish evolution, and changes in the labyrinth may potentially reflect a change from nektonic to near-shore environmental niches. Phylogenetic implications show that endocranial form fails to support monophyly of the 'chirodipterids'. Those with elongated crania similarly fail to form a distinct clade, suggesting these two paraphyletic groups have converged towards either head elongation or truncation driven by non-phylogenetic constraints.

## Editor's evaluation

Clement et al. described and illustrated the endocasts of six Paleozoic lungfish genera from superb 3D fossil material, which are very informative for the understanding of brain evolution of lungfishes, the extant sister group to land vertebrates. Rendering important anatomical details regarding brain evolution in lungfishes and conducting a morphometric analysis, this work will be of interest to a broad evolutionary and paleontological audience.

## Introduction

The field of palaeoneurology was founded a century ago and is a long-established branch of inquiry into fossil vertebrates (*Edinger, 1921*). However relatively little attention has been given to

investigation into the evolution of the brain in fossil fishes, most likely due to the fact that the brain-braincase relationship is generally not as close as in "higher" vertebrates such as birds and mammals (*Jerison, 1973*). Nevertheless, there were some pioneering early works specifically investigating fossil fishes which laid the foundation for further palaeoneurological research in early vertebrates (*Chang, 1982*; *Janvier, 1974*; *Jarvik, 1942*; *Jarvik, 1972*; *Stensiö, 1963*; *Stensiö, 1927*). These days modern advances in imaging technologies are transforming the palaeoneurological landscape, enabling more in-depth investigations of structure-function relationships (*Bruner et al., 2018*; *Walsh and Knoll, 2011*), including in early vertebrates.

Lungfish represent a unique evolutionary lineage, which has persisted for over 400 million years. Their crania differ quite remarkably from other sarcopterygian fishes in having lost the intracranial joint, possession of an autostylic jaw suspension, unique dentition and differences in the number and arrangement of dermal skull bones (*Schultze, 1986*). Homologues of skull roofing bones across sarcopterygians were difficult to identify, leading to the adoption of an alternate (and unique) labelling scheme that continues to be used today (*Forster-Cooper, 1937*), with additions and updates from *Ahlberg, 1991*; *Cloutier, 1997*; *Miles, 1977*. Furthermore, lungfish skulls have undergone considerable reduction in the degree of ossification and the fusion and/or loss of bone, leading to a massive (~10-fold) reduction in the number of dermal skull bones present in extant Dipnoi compared to some Devonian forms (*Criswell, 2015*). However, there has been relatively little attention paid to examining brain evolution in early lungfish (Dipnoi), despite the crania of a large number of fossils having been described since early in the 19th Century (*Sedgwick and Murchison, 1828*), more than 100 fossil lungfish taxa described, and renewed taxonomic interest in lungfish as the extant sister group of the tetrapods.

While aspects of the anatomy of the central nervous system in extant lungfish have been known for close to 150 years (*Collin, 2007*; *Huxley, 1876*; *Northcutt, 1986*), it was only recently that the specific spatial relationship between the brain and the endocranial cavity was investigated. This was examined first in the Australian lungfish, *Neoceratodus* (*Clement et al., 2015*), and later in the lepidosirenid lungfish (*Lepidosiren* and *Protopterus*) and other piscine sarcopterygians (*Challands et al., 2020*). It was found that contrary to earlier reports of the lungfish brain filling just a fraction (10%) of its brain cavity (*Jerison, 1973*), values in fact ranged between 40% and 80% (*Challands et al., 2020*; *Clement et al., 2021*; *Clement et al., 2015*). Moreover, these studies highlighted that the forebrain and labyrinth regions in particular had a close correspondance between brain and endocast, in comparison with the mid and hindbrain regions with a looser association.

Currently, only eight species of fossil lungfish have had the endocast (the negative space within the cranium) described or illustrated in any considerable manner (either full or partial), and are all of Devonian age. These include *Dipnorhynchus sussmilchi* (*Campbell and Barwick, 1982*; *Clement et al., 2016a*), *Chirodipterus wildungensis* (*Säve-Söderbergh, 1952*), '*Chirodipterus*' *australis* (*Henderson and Challands, 2018*; *Miles, 1977*), *Griphognathus whitei* and *Holodipterus gogoensis* (*Miles, 1977*), *Dipnorhynchus kurikae* (*Campbell and Barwick, 2000*), *Rhinodipterus kimberleyensis* (*Clement and Ahlberg, 2014*), and *Dipterus valenciennesi* (*Challands, 2015*). Additional fragmentary neurocranial information is also known in *Scaumenacia curta* (*Boirot and Challands, 2021*).

Some early investigations into fossil lungfish endocranial morphology led *Stensiö, 1963* to suggest that the dipnoan brain 'type' had developed by the beginning of the Devonian and has not undergone any significant changes since that time. However, recent work has shown that the endocasts of some Devonian lungfish continued to change in morphology with the acquisition of new characters certainly into the Upper Devonian period, while also retaining other primitive characteristics (*Challands, 2015*; *Clement and Ahlberg, 2014*; *Clement et al., 2016a*).

The polarity of such characters in the cranial endocasts of fossil lungfish has largely been determined by reference to brains of extant taxa; the Lepidosireniformes (*Protopterus* and *Lepidosiren*) and the Ceratodontiformes (*Neoceratodus*). For example, *Northcutt, 1986*; *Northcutt, 2011* stated that pedunculate olfactory bulbs are the likely primitive state in the Dipnoi as this characteristic is observed in *Neoceratodus*, (considered the most morphologically conservative of the extant dipnoan taxa), and also in the extant coelacanth *Latimeria* (*Dutel et al., 2019*). Furthermore, pedunculate olfactory bulbs are also seen in stem osteichthyans (*Clement et al., 2018*), as well as several more basal outgroups such as placoderms (*Zhu et al., 2021*).

Determining the polarity of endocast characters is important at this juncture in dipnoan palaeoneurological research as we are now in a position to recognise and apply more neurocranial characters to phylogenetic analyses. Following on from Friedman's detailed analysis of lungfish interrelationships based on neurocranial evidence (*Friedman, 2007*), *Clement et al., 2016a* conducted the first phylogenetic analysis of Palaeozoic lungfish based entirely on endocast characters, further emphasising their use for such studies. Despite the problematic homology of dermal bones, endocranial anatomy tends to be far more conserved in vertebrates and thus highly valuable for comparative analysis.

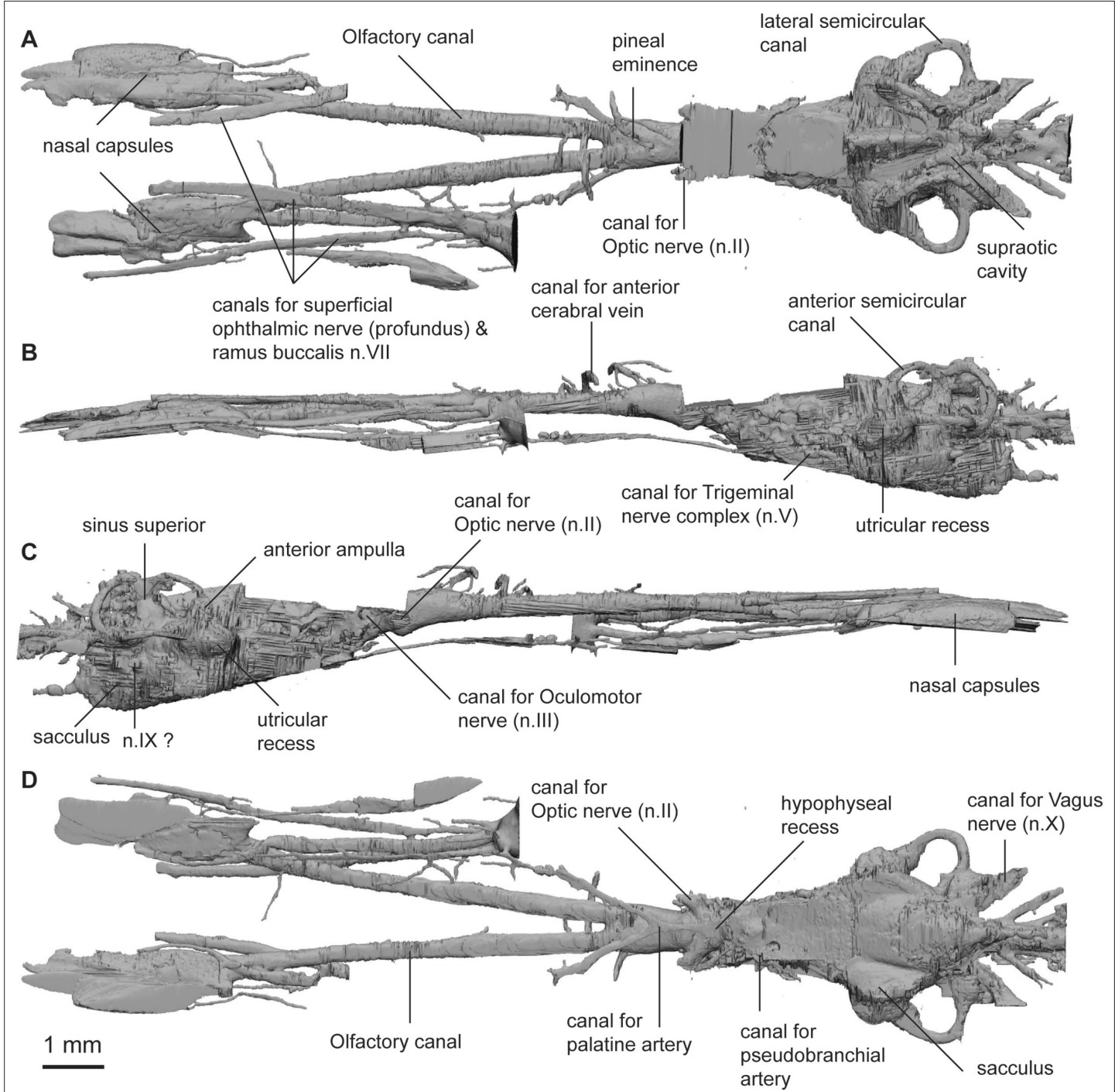

**Figure 1.** Endocast of *Griphognathus whitei* (NHMUK PV P56054) in (**a**), dorsal; (**b, c**), lateral; (**d**), ventral views.

Here, we investigate the endocranial morphology of six Devonian dipnoans (*Iowadipterus halli, Gogodipterus paddyensis, Pillararhynchus longi, Griphognathus whitei, Orlovichthys limnatis,* and *Rhinodipterus ulrichi*) from synchrotron and computed tomography (CT), and compare them with those taxa already known (fossil and extant) using multivariate morphometric analyses (*Figure 1*). We hypothesise that, just as cranial elongation manifests in numerous ways by lengthening of either the snout, jaw or cheek regions, the neurocranium (and by consequence, the endocasts) of Devonian lungfish will have also accommodated elongation by modifying different regions to varying extents.

## Results

### Description of endocasts

#### *Griphognathus whitei* (NHMUK PV P56054)

The skull and endocast of *Griphognathus whitei* show the extremity of the elongated morphotype; the endocast is more than four times longer than it is wide, measuring 130 mm in length but less than 30 mm in width (*Figure 1*). However, it is the olfactory region and forebrain (comprising the olfactory lobes, telencephalon and diencephalon) that have undergone significant elongation, with the mid and hind-brain of similar proportion to that in several other taxa (e.g. *Rhinodipterus* spp.). The shape of the nasal capsules is distinct from other lungfish, in being particularly elongate, forming narrow, oblong ovals. They are situated at the anterior end of remarkably long olfactory canals (comprising almost half the length of the entire endocast) that diverge from each other narrowly at 11°. Numerous canals that may contribute to a medial canal meshwork emanate from the dorsal surface of the olfactory tracts. The lateral branch of the canal for the ramus ophthalmicus profundus n.V and the canal for the ramus buccalis n.VII pass mediodorsally and lateral to the nasal capsule respectively whereas the lateral branch of the nasal vein enters the posterior wall of the nasal capsule.

The telencephalon is elongate with a distinct subpallium. A canal for the anterior cerebral vein is present dorsally on the telencephalon and is distinguished from a pineal eminence by it projecting laterally and curving ventrally. The hypophyseal recess is small and does not protrude far ventrally as in most other lungfish. There are several small canals exiting the hypophyseal recess. To the anterior a single canal exits and then divides representing the canal for the palatine artery. The canal for the pituitary vein is small and exits antero-lateral and slightly dorsal to the canal for the palatine artery. Emanating from the postero-lateral region of the hypophyseal recess the canals for the pseudobranchial arteries are present.

The mesencephalic region of the endocast is not well preserved and its dorsal margin is incomplete, although it appears to have been narrow and short. The rhombencephalic region appears relatively conserved and is similar in proportion to *Rhinodipterus kimberleyensis* (*Clement and Ahlberg, 2014*) of unremarkable length or width, and with space for housing the endolymphatic ducts in the supraoptic cavities dorsally. Unusually, the canals for the trigeminal complex (n.V) do not protrude far laterally in *Griphognathus* and are relatively difficult to discern.

Within the inner ear, the semicircular canals are thin and follow a less circular arc than in *Chirodipterus* and *Gogodipterus*, with the posterior and lateral canals being longer than the anterior canal. The lateral and anterior canals bear prominent swelling (for their ampullae) at their bases, while the ampulla in the posterior canal is less well defined. The sinus superior reaches the same level dorsally as the endolymphatic ducts above the roof of the cranial cavity. The utricular recess is relatively large and oblong when viewed laterally, while the sacculus is not as clearly defined as that in *Rhinodipterus*, and its pouches are separated widely from each other, thereby allowing space for a large notochord.

#### *Iowadipterus halli* (FMNH PF 12323)

Although the resolution of the scan data in *Iowadipterus halli* does not allow for fine detail to be elucidated, overall morphology of the endocast and several other features can nevertheless be recognised (*Figure 2*). The endocast (~21 mm long, 7.5 mm wide) is high and narrow, and closely associated with the notochord ventrally. The telencephalic, diencephalic, and mesencephalic regions are particularly narrow and high, with only a slight widening evident in the hindbrain region.

The nasal capsules are large and oval-shaped, connecting to the greater cranial cavity via their posteromedial corner. The olfactory canals are very short and diverge from each other at 55°. The bulbous appearance of the tracts as they diverge suggest that sessile olfactory bulbs may have been

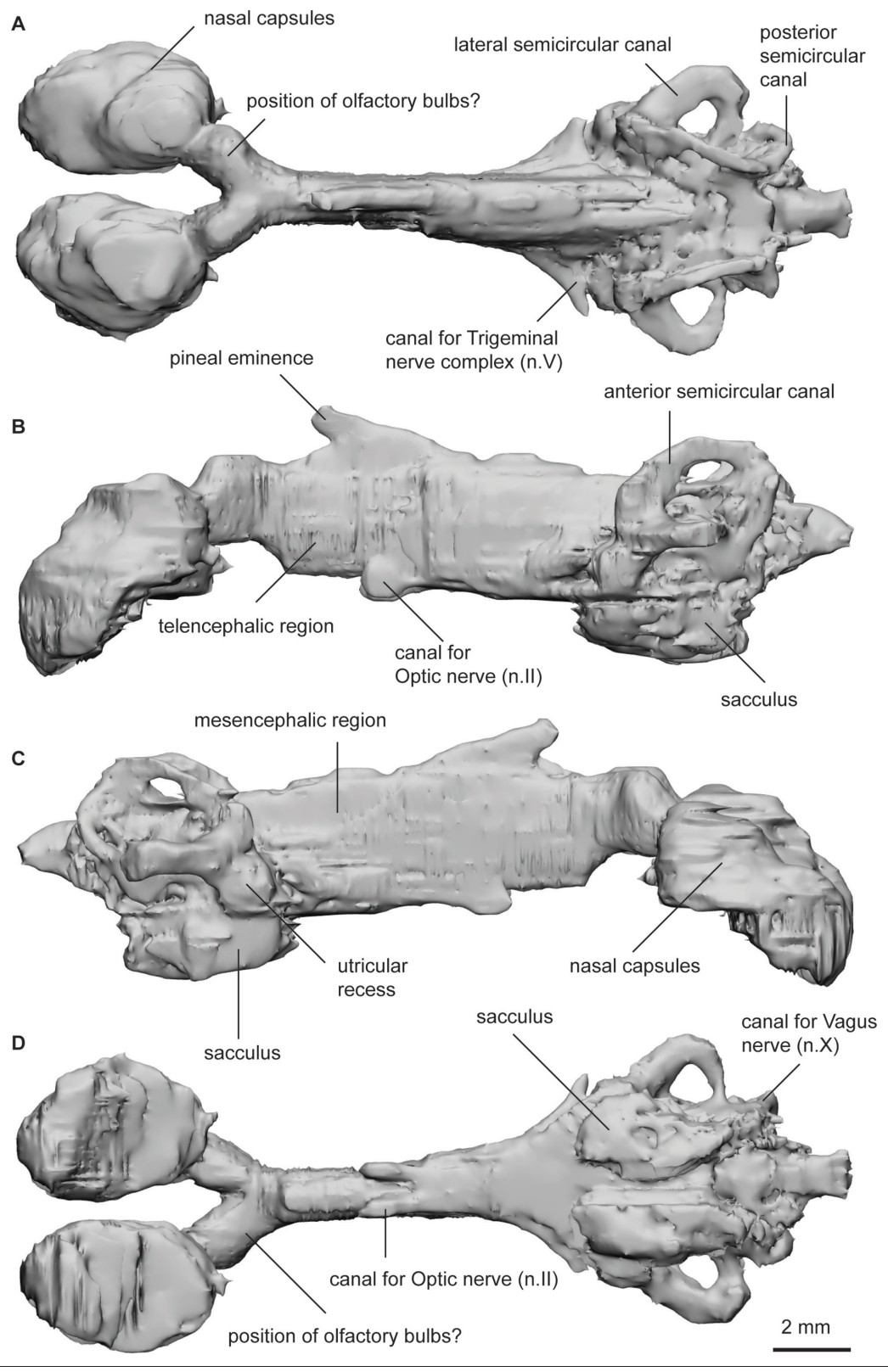

**Figure 2.** Endocast of *Iowadipterus halli* (FMNH PF 12323) in (**a**), dorsal; (**b**), lateral left; (**c**), lateral right; and (**d**), ventral views.

housed here. The telencephalic region has straight dorsal and ventral margins, although ventrally this angles steeply upwards toward the olfactory bulbs suggestive of a slight telencephalic expansion in this region ventrally. An anterodorsally extended eminence on the dorsal surface represents the space for the pineal organ and stalk, and two ventral expansions below this likely housed the optic nerves. The mesencephalic region is similarly high and narrow as is the forebrain.

On either side of the endocast, just anterior of the labyrinths, two laterally extended protrusions probably housed the trigeminal nerve complex. The roof of the cranial cavity is mostly flat and straight in the hindbrain region, and sitting lower than the anterior semicircular canals. A small bulge dorsally in the metencephalic region probably represents the space for the endolymphatic ducts.

The arrangement of the semicircular canals appear somewhat unusual for Devonian lungfish in that the anterior canal is noticeably longer than that of the posterior canal. The anterior semicircular canals are anterior-posteriorly extended, whereas the posterior canals are shorter and are more dorso-ventrally oriented. The lateral semicircular canal has a strong curve and re-enters the labyrinth anterior to the posterior canal. The utricular recess is well-defined on both sides of the specimen, forming almost spherical extensions of the vestibule. Ventral to this, the sacculus forms a single chamber (without a notch to separate the lagena) appearing as an elongated oval-shaped region.

### *Orlovichthys limnatis* (PIN 3725/110)

The endocast of *Orlovichthys limnatis* measures just over 50 mm in length and 20 mm in width (*Figure 3*). The right-hand side is better preserved and includes a complete labyrinth. The nasal capsules (left-hand side preserved) are small and semicircular in outline. The olfactory canals are very short and diverge at 50° from each other. However, overall the forebrain (comprising the olfactory lobes, telencephalon and diencephalon), is extremely long and narrow.

The hypophyseal fossa for the hypothalamus extends posteroventrally and there is only a small protrusion for the pineal eminence dorsally. The canal housing the pseudobranchial artery exits the hypophyseal recess latero-ventrally before abruptly turning postero-laterally. It does not bifurcate into separate canals for the pseudobranchial and internal carotid arteries. In the anterior region of the hypophyseal recess two short canals projecting anteriorly likely housed the palatine arteries. Dorsal to the canals for the pseudobranchial/internal carotid arteries a slender canal projects antero-laterally. This canal held the ophthalmic artery. Just posterior to this canal, the canal for the pituitary veins project laterally and slight dorsally.

The dorsal margins of the telen-, dien-, and mesencephalon regions are incomplete but appear to have been relatively flat. The optic nerve canals (n.II) exit antero-laterally from just behind the pronounced subpallium of the telencephalon. Slightly dorsal and posterior to n.II is a small canal that exits perpendicular to the lateral wall of the mesencephalon and represents the canal for the oculo-motor nerve. A canal for the trochlear nerve is not visible. The trigeminal nerve complex exits the cranial cavity branching into two separate canals extending anterolaterally. *O. limnatis* lacks highly expanded space to house the endolymphatic ducts as is present in some other taxa (e.g. '*C*'. *australis*, *R. kimberleysensis*), but the posterior extent of the supraoptic cavities is visible. A canal for the abducens nerve (n.VI) projects anteriorly from the anterior of the saccule.

The labyrinth system comprises three semicircular canals, each bearing prominent, elongate ampullae and forming small, circular arcs. The point where the anterior and posterior semicircular canals join, the crus commune, sits above the roof of the cranial cavity. *O. limnatis* has a prominent, oval-shaped utriculus and an elongated sacculus pouch, which is inferred to have been large although it is incomplete ventrally.

### *Pillararhynchus longi* (ANU 49196) - with additional notes on *Gogodipterus paddyensis* (WAM 70.4.250)

Due to their similarity and incompleteness of *Gogodipterus*, the following description is based predominantly on *Pillararhynchus*, but where features differ in *Gogodipterus* further remarks are included. *Gogodipterus paddyensis* was previously described as *Chirodipterus paddyensis* (***Miles, 1977***), before the material was further acid-prepared and redescribed as Gogodipterus by *Gogodipterus* by ***Long, 1992***. Only the inner ear canals and part of the trigeminal nerve complex are preserved as it is missing a large part of the dorsal and anterodorsal faces of the endocranium.

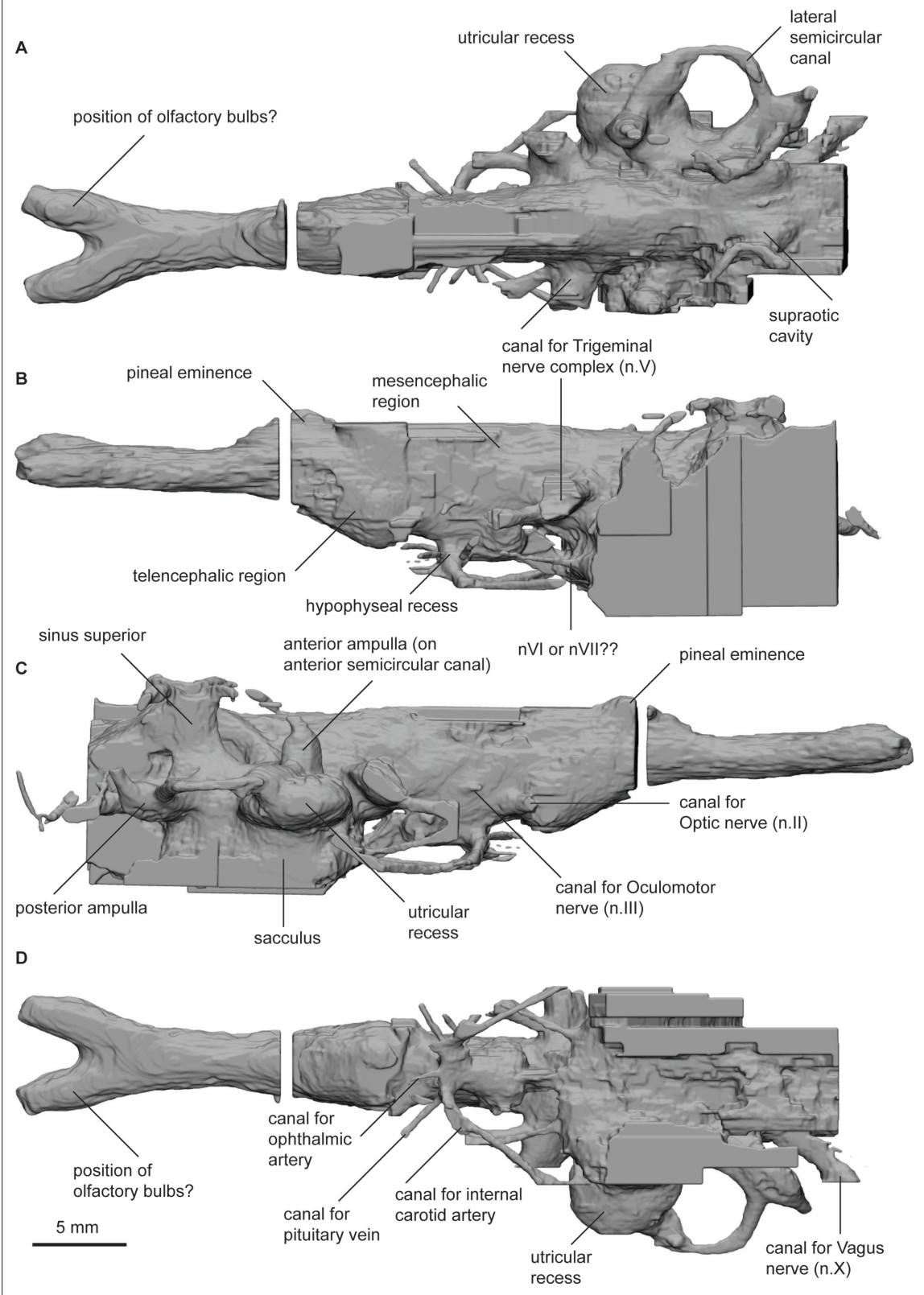

**Figure 3.** Endocast of *Orlovichthys limnatis* (PIN 3725/110) in (**a**), dorsal; (**b**), lateral left; (**c**), lateral right; and (**d**), ventral views.

As noted in *Barwick and Campbell, 1996*, *Pillararhynchus longi* (ANU 49196) is seemingly a small and was presumably a young individual with the following measurements: skull 46 mm long, 33 mm wide and ~31 mm high (*Figure 4*). The braincase is well preserved but most of the skull roof is missing. The endocast itself measures 38 mm in length. In comparison, *Gogodipterus paddyensis* (WAM 70.4.250) was a larger individual and although incomplete anteriorly, had a skull at least 75 mm in width (*Figure 5*). The holotype is also lacking dermal skull bones, but the oticoccipital region of the braincase is well preserved and allows for comparison of this region with other taxa.

The nasal capsules are large, oblong, with a convex dorsal surface and remain open ventrally. There are no discernible divisions within the two capsules. The large canals for the olfactory nerves (n.I) enter the nasal sacs in their posteromediodorsal region. Below these nerves, there are two other canals exiting the nasal capsule posterolaterally; the larger canal likely contained the orbitonasal vein, and smaller canal likely contained the palatine nerve (VII). The olfactory canals diverge at around 45° from each other. They remain relatively narrow except for a slight bulge just anterior of where they join the telencephalic region, suggesting that these swellings housed sessile olfactory bulbs. On the right side, there is a single canal exiting dorsally from one of these swellings, before changing course to an anterolateral direction. This may represent a branch of the anterior cerebral vein.

The forebrain region is narrow but dorsoventrally raised. It is dominated by a large single recess for the pineal organ, extending anterodorsally over the junction of the olfactory canals and the telencephalon. In line with the pineal eminence, but situated ventrally, are the two large olfactory canals. Anterior to these in ventral view, a slight swelling for the telencephalon can be seen. In the diencephalic region the hypophyseal recess extends directly ventrally beneath the cranial cavity. Two pairs of canals exit the hypophysis; furthest ventrally and extending anterolaterally is the canal for the ophthalmic artery, and those extending directly laterally carried the pituitary vein.

The midbrain is wider than the preceding forebrain but much shallower. The dome for the mesencephalon represents the highest dorsal extent of the cranial cavity. On the left side of the specimen, a small canal situated midway up the cranial cavity wall would have housed the oculomotor nerve (n.III), and further dorsal to this is another opening, which may represent the remains of the trochlear canal (n.IV).

The anterior metencephalic region is marked by a large bifurcating canal for the trigeminal nerves (n.$V_1$ and n.$V_{2\&3}$) projecting laterally. Just posterodorsal to these are the similarly large canals for the facial nerves (n.VII), which soon reconnect with n.$V_{2\&3}$. The dorsal extent of the rhombencephalon is flat and broad, and significantly lower than that of the mesencephalon and labyrinth with the exception of the two rounded protuberances of the supraotic cavities. There are no obvious endolymphatic ducts in *P.longi*, although one is present on the right side in *G. paddyensis*. The roof of the rhombencephalic portion of the cranial cavity in *G. paddyensis* is also not as flat as that of *P.longi*. Two large canals posterior to the labyrinths house the vagus nerves (n.X), and several other smaller canals in this region would have carried smaller spinal nerves. Ventral to the cavity for the spinal cord and separated by bone is the large notochordal canal, although the anterior-most extent of the canal cannot be gleaned from the scan data due to very bright pyritic inclusions in the specimen obscuring this and the form of the sacculolagenar region.

The labyrinth region on the right side is well preserved, containing all three semicircular canals and the utricular recess. Unfortunately, the boundaries of the sacculolagenar pouch are not observable and thus its shape cannot be reliably determined. The semicircular canals are robust and form small circular arcs; although the arcs in *P. longi* are slightly more ovoid than those of *G. paddyensis*. Each canal possesses a sizeable elongate expansion (ampulla) at its base. The crus commune is situated ventral to the highest points of the anterior and posterior semicircular canals, and the sinus superior is also prominent above the roof of the cranial cavity. The posterior canal is the longest of the three semicircular canals, joining with and at the level of the lateral canal, rather than inserting on top of it as the anterior semicircular canal. The lateral canal is the shortest of the three semicircular canals. Although the sacculolagenar region is not preserved, by comparison with the semicircular canals, the spherical utricular recess appears relatively small. The semicircular canals of *G. paddyensis* are well preserved but almost indistinguishable from those in *P. longi*.

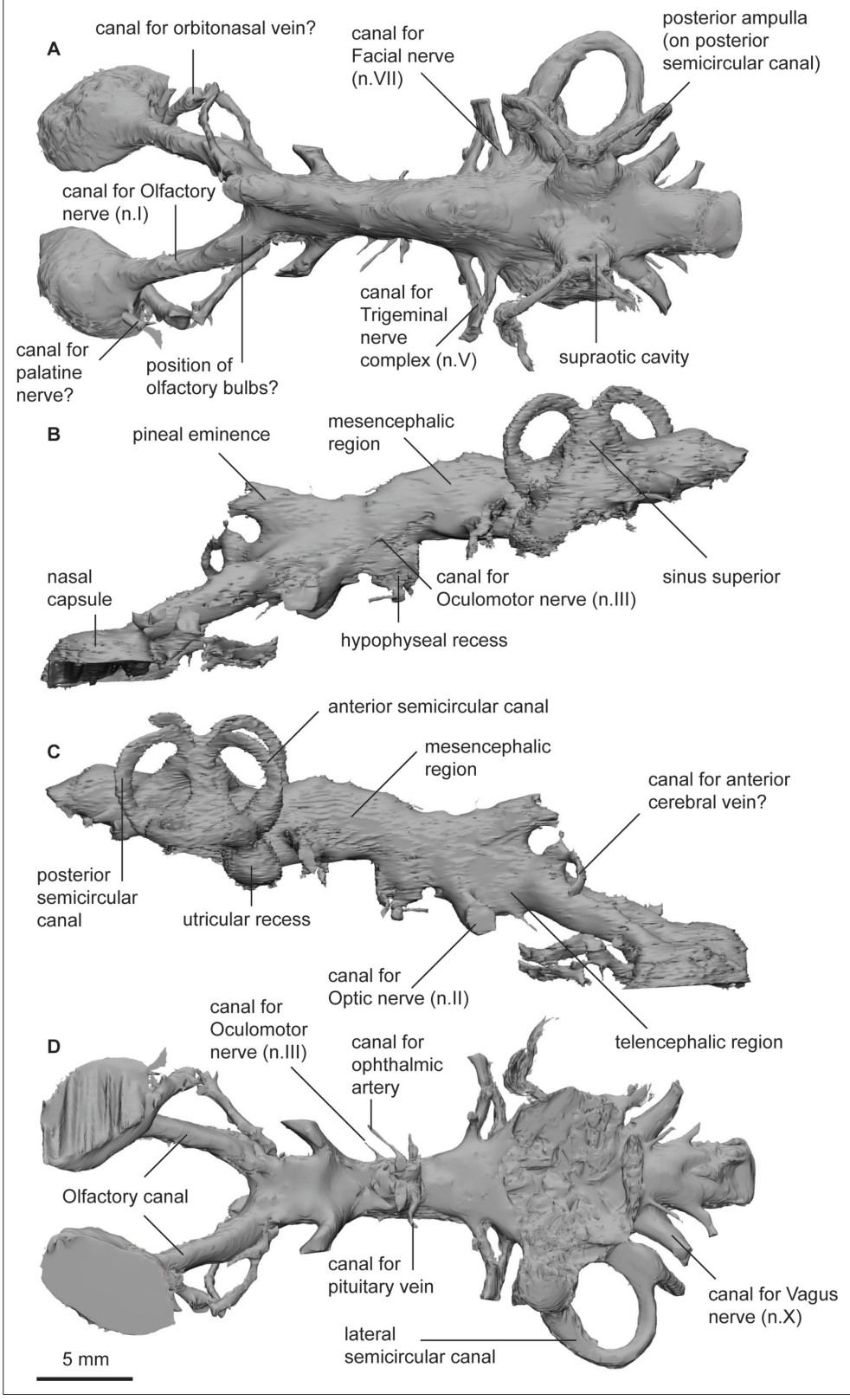

**Figure 4.** Endocast of *Pillararhynchus longi* (ANU 49196) in (**a**), dorsal; (**b**), lateral left; (**c**), lateral right; and (**d**), ventral views.

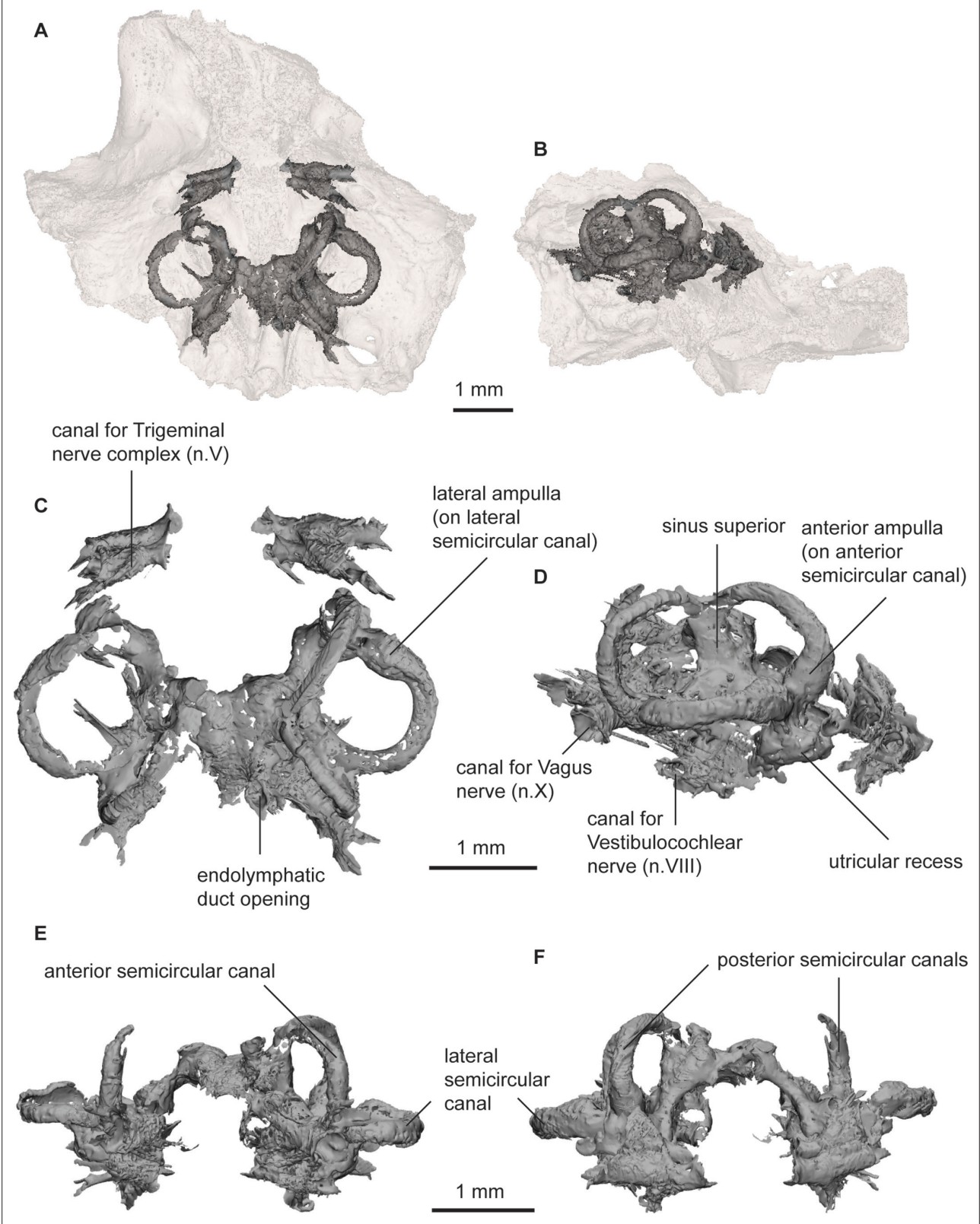

**Figure 5.** Endocast of *Gogodipterus paddyensis* (WAM 70.4.250) labyrinth region in (**a**), dorsal; and (**b**), right lateral views with transparent skull overlay; (**c**), dorsal; (**d**), right lateral; (**e**), anterior; (**f**), posterior views of labyrinth endocast only.

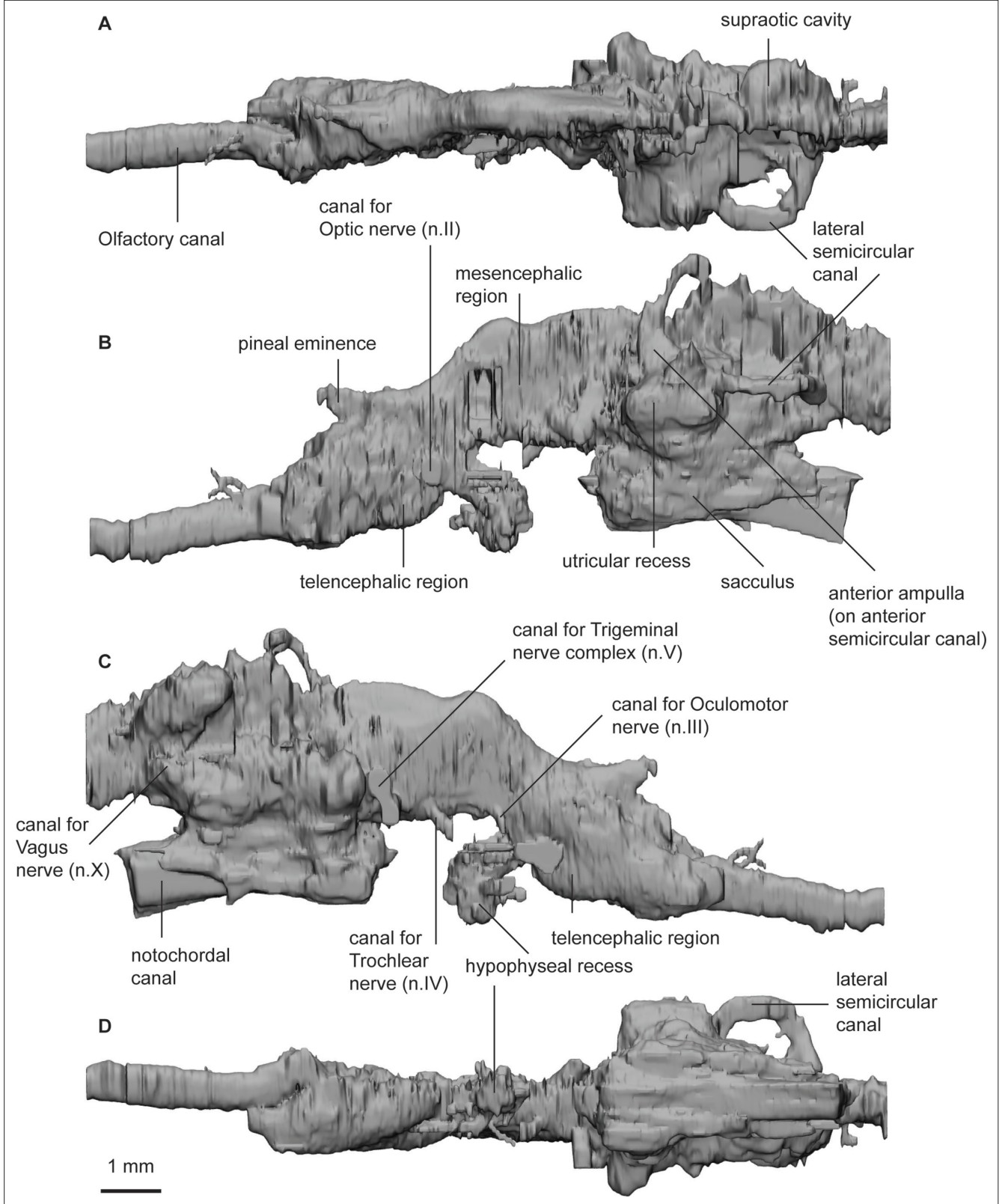

**Figure 6.** Endocast of *Rhinodipterus ulrichi* (NRM P6609a) in (**a**), dorsal; (**b**), lateral left; (**c**), lateral right; and (**d**), ventral views.

### *Rhinodipterus ulrichi* (NRM P6609a)

The endocast of *Rhinodipterus ulrichi* is incomplete anteriorly without the nasal capsules preserved (***Figure 6***). Overall, the endocast is extremely narrow (14 mm in length and would have been only about 5 mm wide in life), although the specimen has undergone some lateral compression. Only the left olfactory canal is preserved so the angle of divergence between them is not known, although it appears to have been very small.

The long and narrow olfactory canal runs directly anteroposteriorly before reaching the telencephalon which bears a prominent ventral bulge and a steeply posterodorsally angled dorsal margin (which continues to rise all the way up to the mesencephalic portion of the endocast). There is a large pineal eminence projecting anterodorsally from the roof of the diencephalic region. The hypophyseal fossa is of moderate size and extends posteroventrally from the lower margin of the endocast. The large, circular canals for the optic nerves (n.II) are visible close to the ventral boundary between the telencephalic and diencephalic regions. Smaller canals that would house nerves n.III (oculomotor) and n.IV (trochlear) exit the cranial cavity close to the ventral margin of the mesencephalic region.

The mesencephalic region is unusual in being the highest point of the endocast with a dorsal bulge in this area. The divergence between the rostral forebrain and caudal hindbrain is large at 140° and seemingly unusual in lungfish with the condition not known in any other taxon.

On the right side of the specimen, it appears there may have been space for enlarged supraoptic cavities in *Rhinodipterus ulrichi*, similar to the condition in *R. kimberleyensis*. The dorsal margin of the hindbrain is similarly domed as is the midbrain, although not as distinct. The bifurcating canal for the trigeminal complex is best visible on the right-hand side, as is the canal posteriorly for the vagus nerve (n.X).

The anterior and lateral semicircular canals are preserved on the left side, but only the anterior canal appears to bear an ampullar swelling, somewhat triangular in shape, where it connects with the utriculus. The utricular recess is relatively large and is somewhat lemniscate in outline when viewed laterally. The sacculolagenar pouch is elongate and tapers posteriorly. The notochordal canal is preserved between the two sacculolagenar pouches and extends anterior to the level of the trigeminal nerve complex.

## Principal Component Analyses

Firstly, preliminary standard Principal Component Analyses (PCAs) were performed on matrices of various measurements taken from the endocasts (***Figure 7***). These matrices were partial in order to maximize either the number of variables (PCA-characters; 13 variables on seven taxa) or the number of taxa (PCA-taxa; nine variables on nine taxa). Imputation (i.e., methods to fill in or impute missing data in a matrix) has not been used for these standard PCA. In both analyses, the most incomplete taxa (i.e., *Griphognathus whitei, Protopterus aethiopicus, Gogodipterus paddyensis, Dipnorhynchus sussmilchi,* and *Rhinodipterus ulrichi*) were excluded, thus results from these analyses do not encompass the whole spectrum of disparity. Furthermore, in both analyses, three variables (length and height of the sacculus, length of nasal capsules) were not included because of the high proportion of incomplete data. Based on the broken-stick method, only PC1 was non-trivial for the PCA-taxa, whereas the first three axes were considered as non-trivial for the PCA-characters.

In the PCA (taxa), variables associated with the semicircular canals (height, length, and width) and variables taking into consideration nerves (n.II, n.V, n.X) position covaried together and represent 69.9% of the total variation (***Figure 8***, ***Figure 8—figure supplement 1***). Although considered as trivial based on the broken-stick method, the second axis reflects a contrast between the length of the olfactory canals and the angle between these canals and it accounts for 17.1% of the total variation; this important source of variation is important even without the inclusion of *Griphognathus*.

In the PCA (characters), all variables, with the exception of the angle between the olfactory canals, are increasing simultaneously accounting for 55.2% of the total variation (***Figure 9***, ***Figure 9—figure supplements 1 and 2***), whereas the variation associated with the angle between the olfactory lobes and height of n.II foramen accounts for the variation of PC2 (16.4%). This tendency is even clearer in the projection of axes PC1 *versus* PC3 (***Figure 9B***). In addition, there is a contrast between the utriculus (height and length) and the semicircular canals (height, length, and width). The height and depth of n.V are opposed to the distance between n.V and n.X.

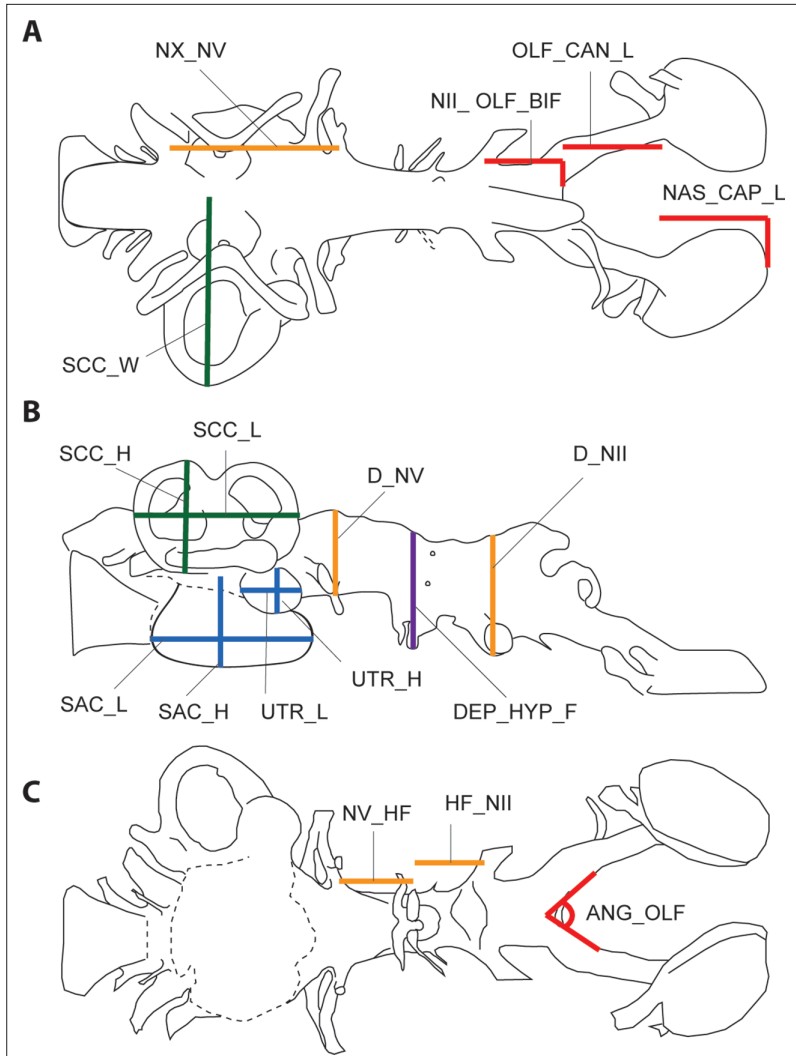

**Figure 7.** Illustration of 17 endocast variables measured on endocast in (**a**), dorsal; (**b**), left lateral; and (**c**), ventral view. ANG_OLF, angle of bifurcation between olfactory canals; D_NII, depth of cranial endocast at level of n.II; D_NV, depth of cranial endocast at level of n.V; DEP_HYP_F, depth of cranial endocast at level of hypophyseal fossa; HF_NII, distance from hypophyseal fossa to n.II; NAS_CAP_L, length of nasal capsules; NII_OLF_BIF, distance from n.II to point of bifurcation between olfactory canals; NV_HF, distance from n.V to hypophyseal fossa; NX_NV, distance from n.V to n.X; OLF_CAN_L, length of olfactory canals from nasal capsules to point of bifurcation; SAC_H, height of sacculus; SAC_L, length of sacculus; SCC_H, height of semicircular canals; SCC_L, length of semicircular canals from anterior to posterior; SCC_W, width of semicircular canals from midline to lateral canal; UTR_H, height of utriculus; UTR_L, length of utriculus.

In both standard PCAs (PCA-taxa and PCA-characters), some variables cluster together [(1) the three variables (length, height, and width) of the semicircular canals and (2) the depth at nerves II and V] and the angle between the olfactory lobes show a clear distinction from the remaining variables. Although some trends are common to both standard PCAs, differences of results between the PCA (taxa) and the PCA (characters) highlight the advantages of using other methods such as Bayesian PCA (BPCA) and the PCA for incomplete data (InDaPCA). These two methods allow us to perform more integrative analyses using distinct approaches to deal with incomplete data. Furthermore, the most incomplete taxa (*Griphognathus, Dipnorhynchus, Gogodipterus,* and *Protopterus*) and variables (length of nasal capsules and the sacculus length and height) can be included in the analyses.

Thus, using Bayesian Principle Component Analysis (BPCA), only three axes are considered to be statistically non-trivial based on the broken-stick method (*Figure 10*, *Figure 10—figure supplement 1*). The first axis accounts for 54.2% of the variation, while the second axis explains 29.2%. The main

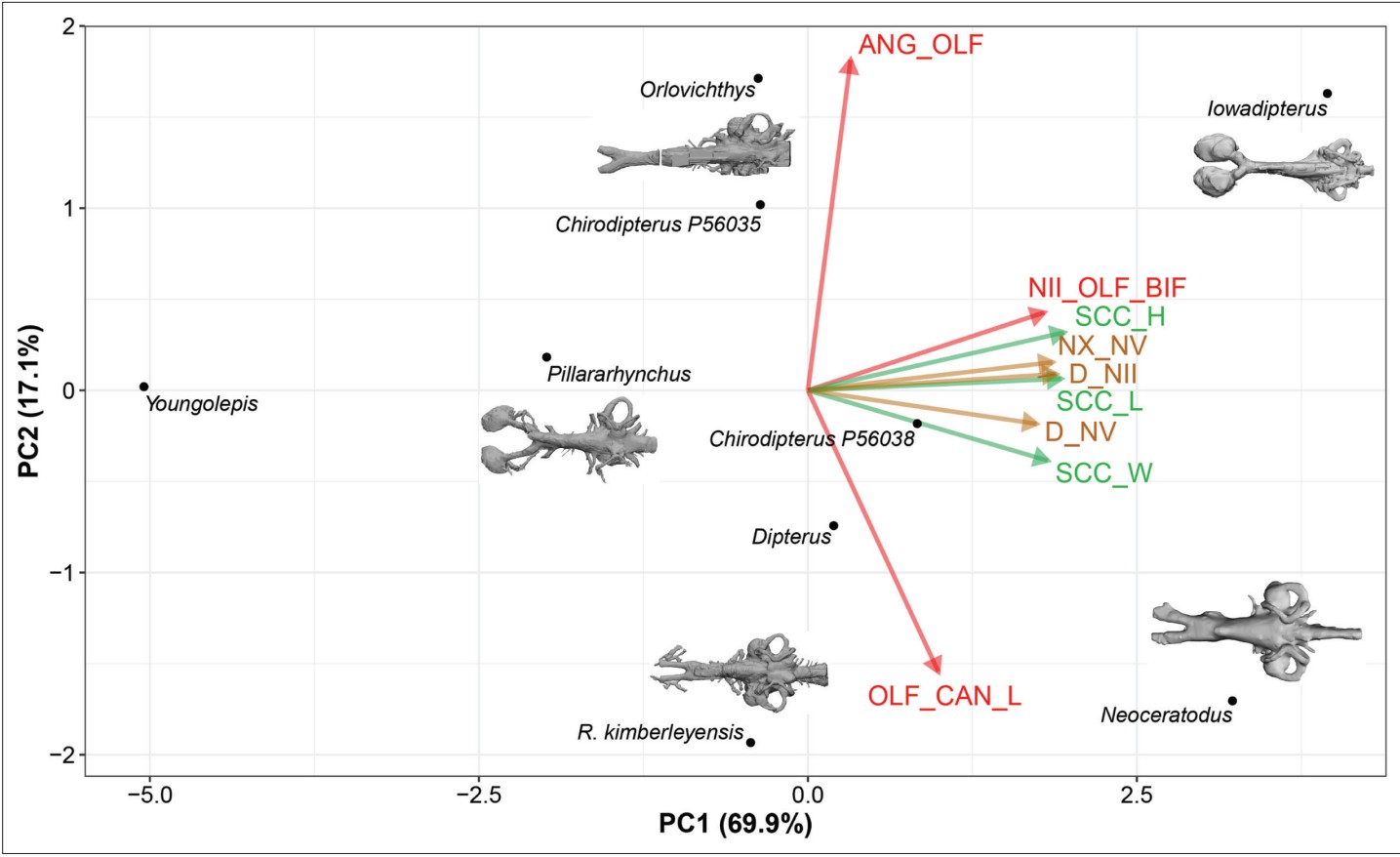

**Figure 8.** Standard Principal Component Analysis (PCA) biplot of lungfish endocast shape optimised for maximum number of included TAXA, PC1 vs PC2. Loadings and eigenvalues shown in *Figure 8—figure supplement 1*.

The online version of this article includes the following figure supplement(s) for figure 8:

**Figure supplement 1.** Standard PCA loadings optimised for TAXA A, PC1; and B, PC2.

source of variation of the first axis is a contrast between the length of the nasal capsules and the olfactory canals (*Figure 10b–c*), and the angle between the olfactory canals. All the remaining variables respond similarly (all change concordantly with the tandem length of the nasal capsules – length of the olfactory canals) corresponding to a minor source of variation for axes 1 and 2. In summary, BPCA PC1 and PC2 summarize the general endocranial tendency, whereby the length of the nasal capsule +olfactory canals increase as the angle between the olfactory canals decreases. *Griphognathus whitei* has a particularly strong weight on this tendency (*Figure 10a*). The three variables associated with the semicircular canals (length, height, and width) cluster together, as do the height and length of the sacculus, the height and length of the utriculus and the depth of n.II and n.III.

Further findings worthy of note include *Gogodipterus paddyensis* failing to cluster near the rest of the 'chirodipterids', instead plotting out closer to *Dipnorhynchus sussmilchi*. The extant taxa, *Neoceratodus* and *Protopterus* do not share a similar morphospace, whereas *Youngolepis praecursor*, *Chirodipterus australis*, *Dipterus valenciennesi*, *Orlovichthys limnatis*, *Pillarhynchus longi*, and *Rhinodipterus* spp. do (*Figure 10a*).

Finally, using PCA for incomplete data (InDaPCA), the first axis represents 66.0% of the variation, while the second and third axes represent 12.6% and 6.46%, of the variation, respectively (*Figure 11*, *Figure 11—figure supplement 1*). Based on the broken-stick method, solely axis 1 is non-trivial, but the interpretation of the first three axes is biologically meaningful (*Figure 11—figure supplement 1*). The main difference between the scatter diagrams of the InDaPCA and the BPCA is that the taxa in the InDaPCA morphospace occupy a better space distribution where the functional out-group *Youngolepis* is clearly distinct from other lungfish on the first axis.

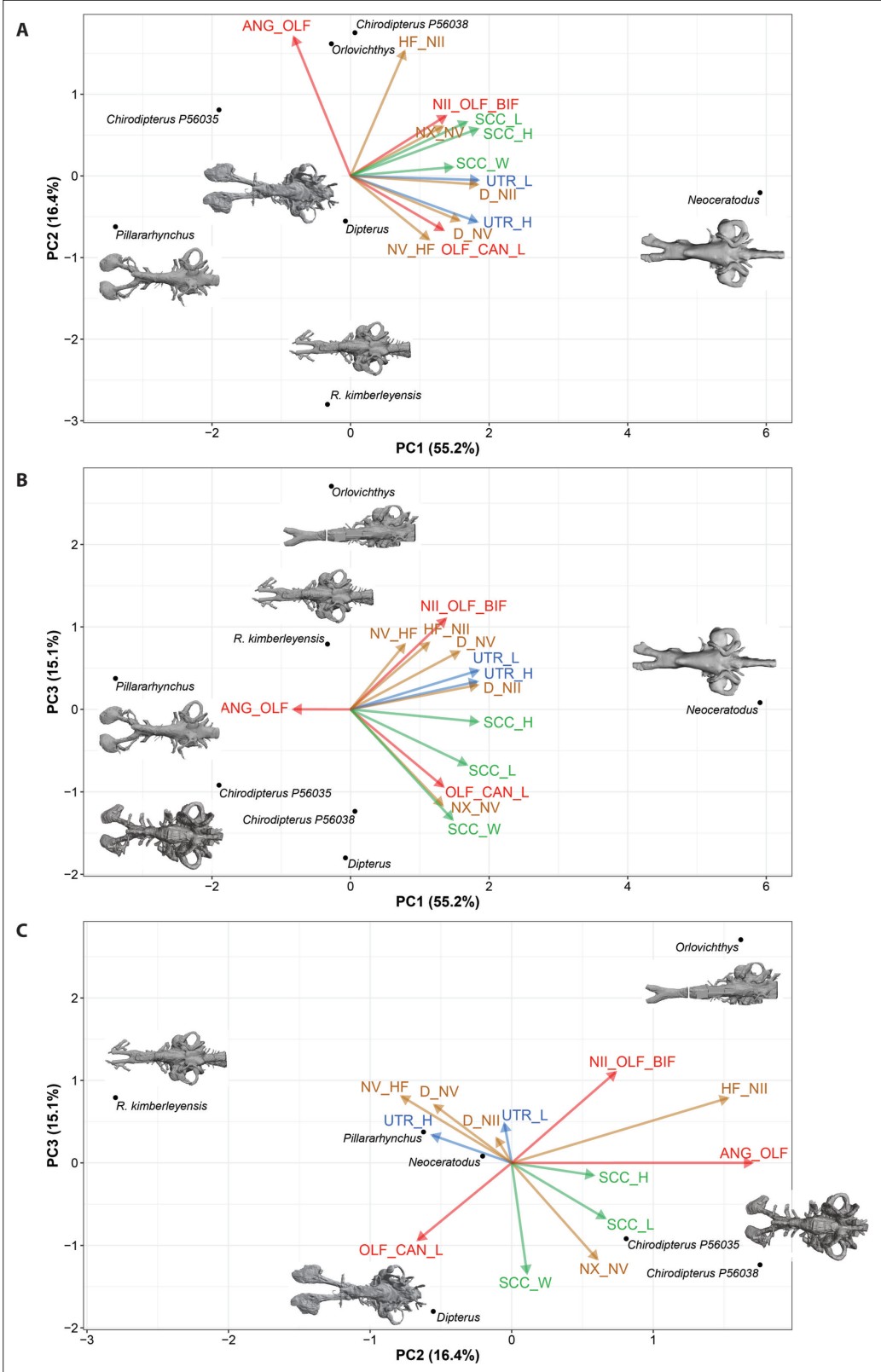

**Figure 9.** Standard Principal Component Analysis (PCA) biplot of lungfish endocast shape optimised for maximum number of included CHARACTERS. (**A**), PC1 vs PC2; (**B**), PC1 vs PC3; (**C**), PC2 vs PC3. Loadings and eigenvalues shown in *Figure 9—figure supplements 1 and 2*.

*Figure 9 continued on next page*

*Figure 9 continued*

The online version of this article includes the following figure supplement(s) for figure 9:

**Figure supplement 1.** Standard PCA loadings optimised for CHARACTERS A, PC1; B, PC2; and C, PC3.

**Figure supplement 2.** Standard PCA eigenvalues optimised for maximum A, CHARACTERS; and B, TAXA.

The first axis in the InDaPCA corresponds to an integration of all variables with the exception of the angle between the olfactory canals (*Figure 11—figure supplement 1*), which has a weak importance on the first axis. This main source of variation corresponds to a coordinate integration of the endocast components. Axis PC2 reflects mainly the contrasts between (1) the length of the olfactory canal and the angle between these canals. The size and shape of the sacculus and semicircular canals, the size of the utriculus, and the depth at cranial n.II and n.V and the height of the foramen for cranial n.II form three clusters in the projection of axes 1 and 2. Projections of axes PC1 *versus* PC2, and PC1 *versus* PC3 provide similar endocranial information.

For all the different projections some variables consistently cluster together: (1) the three variables (length, height, and width) of the semicircular canals, (2) the height and length of the sacculus, (3) the height and length of the utriculus and to a lesser extent (4) the depth at cranial nerves II and V. The projection of axes PC2 and PC3, which accounts for 19.36% of the variation, permits a better visualization of the integration and contrasts among variables. On axes PC2 and PC3, the height of the foramen for cranial nerve II, the distance between cranial nerve II and the olfactory bifurcation and the length of the nasal capsule do not contribute to the main source of variation. On the projection of axes PC2 *versus* PC3, there is a strong signal that the sacculus is better integrated with the semicircular canals and opposed to the utriculus. Thus, when the parameters associated with the sacculus and semicircular canals increase the parameters for the utriculus decrease. Although when these parameters are combined it is part of a complex system, these results suggest a modular dissociation. The main contrasts on this projection correspond to (1) the angle between the olfactory canals *versus* the length of these canals, and (2) the distance between cranial n.V and n.X and the semicircular canal parameters *versus* the depth at cranial n.II and n.V and the height of the foramen of cranial n.V.

In contrast to the BPCA, using InDaPCA results in *Gogodipterus paddyensis* clustering near the rest of the 'chirodipterids'. The extant taxa, *Neoceratodus* and *Protopterus* do not share a similar morphospace, whereas *Chirodipterus australis*, *Dipterus valenciennesi*, *Orlovichthys limnatis*, and *Pillarhynchus longi* cluster close to one another.

## Discussion

Lungfish first appear in the fossil record in the Early Devonian as robust marine animals, many with hyper-mineralised tooth plates, heavily-ossified dermal skulls and neurocrania (*Cui et al., 2022*). However, from the Late Devonian-Carboniferous onwards, many undergo a decline in the degree of ossification as their skeleton becomes more cartilaginous. This reduction in the dipnoan cranial dermal skeleton was likely accompanied by some changes in the cranial cavity form, but this level of increasingly poor ossification of the neurocranium in those taxa limits our possible comparisons to between the more heavily ossified Devonian forms and extant taxa only. Consequently, some of the oldest lungfish taxa are the most completely known, especially with respect to neurocranial morphology. With the exception of the Permian lungfish *Persephonichthys chthonica* (*Pardo et al., 2014*), no post-Devonian fossil taxa have had their cranial endocasts described. We acknowledge that our investigation of lungfish brain evolution as elucidated from morphometric analysis of cranial endocasts is still preliminary in several respects. Due to the extreme age and rarity of 3D preserved lungfish endocrania, we used all material available to us. Admittedly we cannot hope to control for many factors, including ontogeny, so our interpretations must naturally take that into consideration. However, we hope that our study inspires future work on the neural evolution of both fossil and extant lungfish.

### Implications for sensory abilities in early lungfish

Overall, there is considerable variation in the shape and relative proportions of the endocasts examined. *Griphognathus whitei* stands out as the most extreme case of elongation of the olfactory region, whereas other taxa, such as *Iowadipterus halli*, appear to elongate the midbrain region,

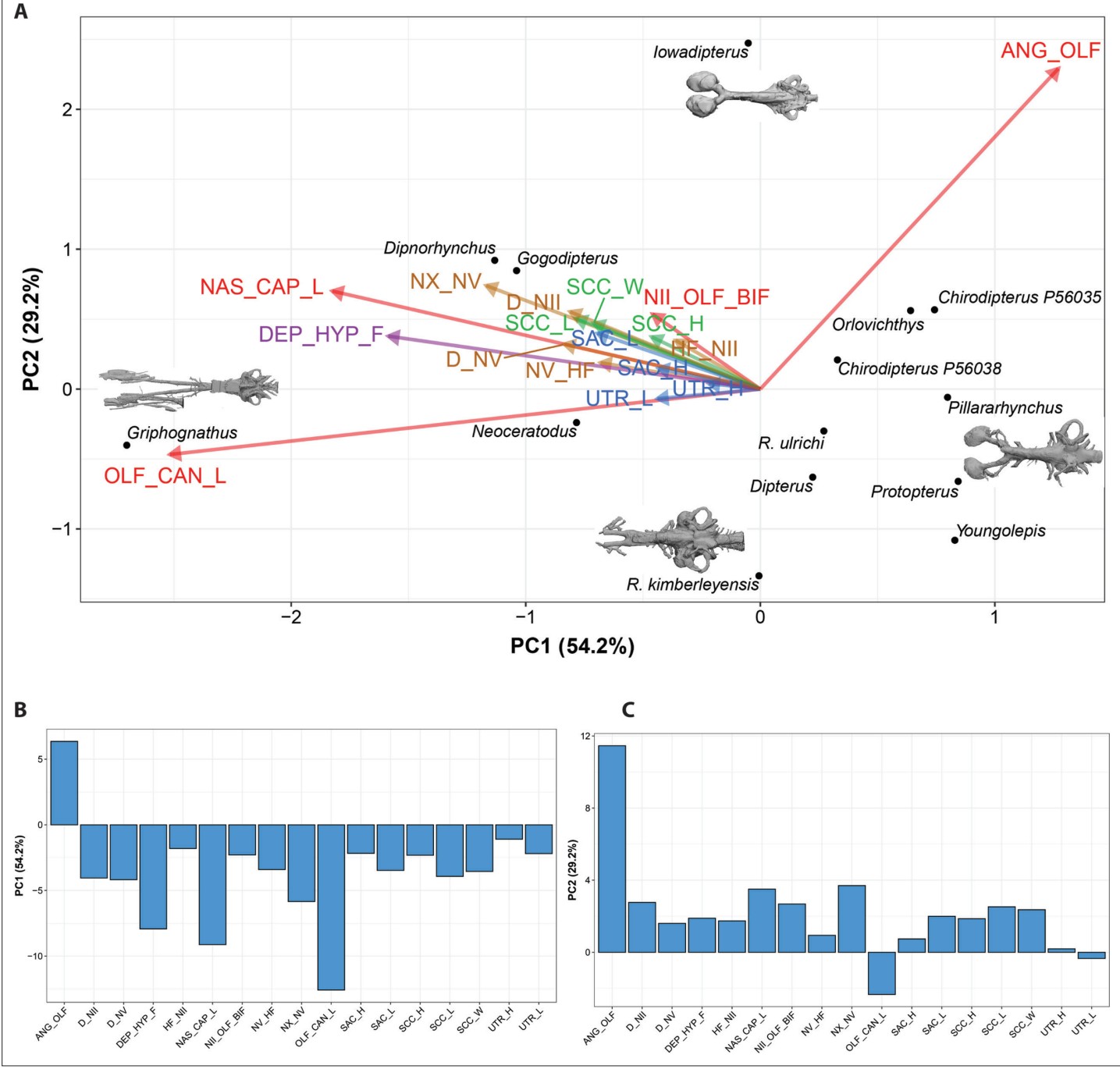

**Figure 10.** Bayesian Principle Component Analysis (BPCA) biplot of lungfish endocast shape (**A**), PC 1 vs PC 2 with endocasts of select taxa shown; and (**B**), (**C**) loadings for PCA 1–2. Eigenvalues shown in *Figure 10—figure supplement 1*.

The online version of this article includes the following figure supplement(s) for figure 10:

**Figure supplement 1.** PCA eigenvalues for A, Bayesian Principle Component Analysis (BPCA); and B, Principle Component Analysis for incomplete data (InDaPCA).

whilst retaining very short olfactory canals and more conventionally-shaped nasal capsules (rounded to slightly oval). Although only a partial endocast, *Gogodipterus paddyensis* is nearly indistinguishable from *Pillararhynchus longi* in labyrinth morphology. All 'chirodipterid' taxa (sensu *Friedman, 2007*) appear to share mediolaterally-oriented nasal capsules, close to equidimensional circular arcs of their semicircular canals and a broad angle of bifurcation between the olfactory canals. Despite this, the

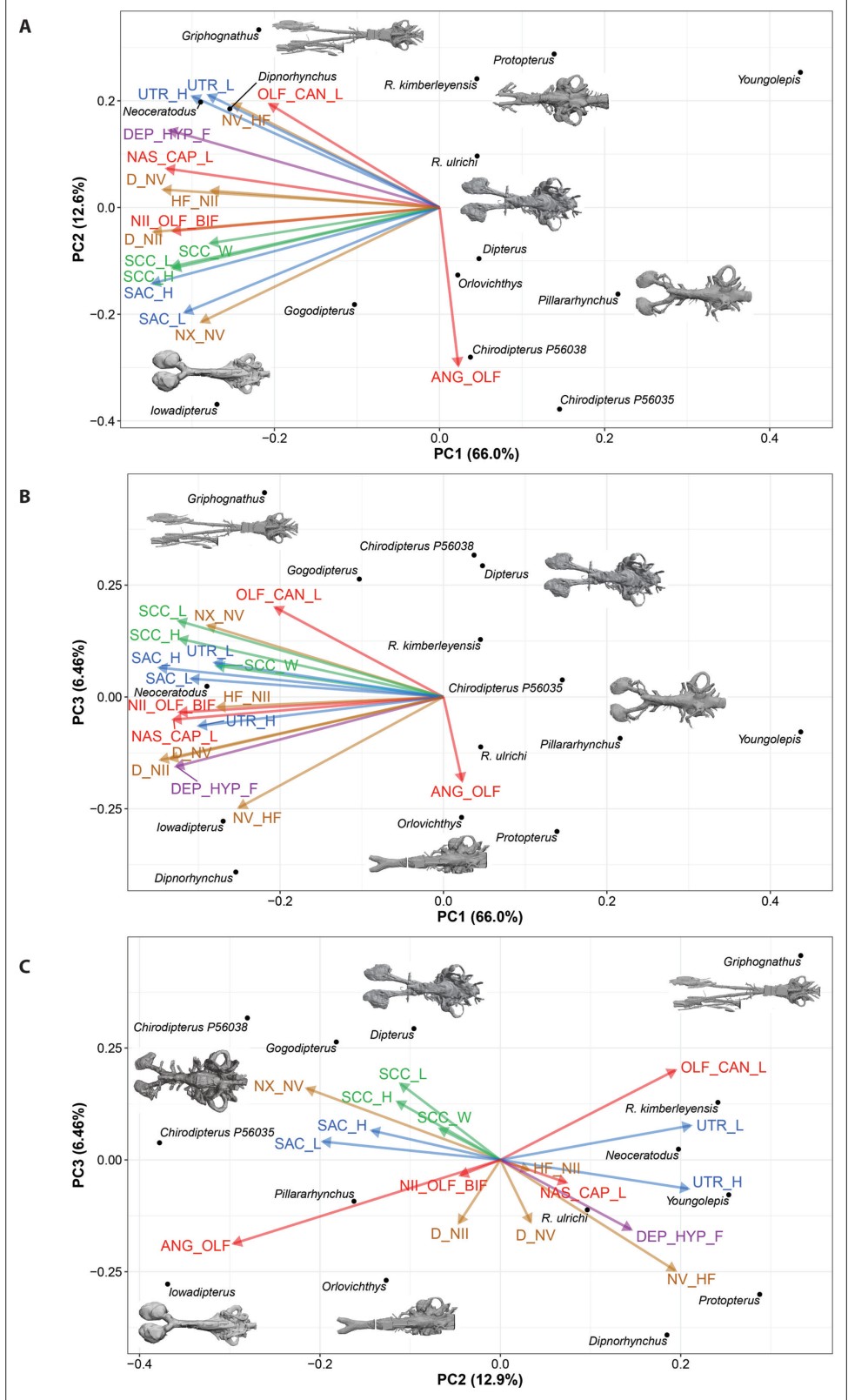

**Figure 11.** Principle Component Analysis for incomplete data (InDaPCA) biplot of lungfish endocast shape for (**A**), PC1 vs PC2; (**B**), PC1 vs PC3; (**C**), PC2 vs PC3, with endocasts of select taxa shown. Explanation of variable abbreviations as per **Figure 7**.

*Figure 11 continued on next page*

*Figure 11 continued*

The online version of this article includes the following figure supplement(s) for figure 11:

**Figure supplement 1.** Principle Component Analysis for incomplete data (InDaPCA) loadings for A, PC1; B, PC2; and C, PC3.

'chirodipterid' taxa do not consistently cluster together in the analyses, giving further weight to the proposition that these taxa form a paraphyletic group (*Friedman, 2007*).

*Rhinodipterus ulrichi* and '*Chirodipterus*' *australis* have endocasts with strongly anteroventrally-directed fore/midbrain portions in comparison with the hindbrain, while presumed close relatives *R. kimberleyensis* and other 'chirodipterid' taxa do not display this character. Aside from this, the endocasts of the two *Rhinodipterus* species are broadly similar to each other – the absence of an eminence for the pineal organ in *R. kimberleyensis* is likely simply a preservational artefact (*Clement and Ahlberg, 2014*). The hypophyseal recess varies from being directed ventrally (*R. kimberleyensis, Orlovichthys limnatis, P. longi*) or posteroventrally (*Dipnorhynchus* spp., *Dipterus valenciennesi, R. ulrichi*).

The length of the olfactory canals (usually incorporating olfactory nerve, bulb and tract) and their angle of bifurcation are one of the largest sources of variation among endocasts of different lungfish taxa (*Figure 8*), and unsurprisingly, these two features are strongly linked. The shape of the olfactory capsules appears highly variable in Devonian lungfish endocasts (*Figure 12*). These are most commonly oval-shaped, although the orientation can vary. Others are highly elongate as in *Griphognathus whitei*, to triangular in '*Chirodipterus*' *australis*. Our study simply captured length but future work on shape and orientation may prove pertinent to the overall plasticity of the olfactory region. The other strongest signals from our data concern the labyrinths, specifically the size and shape of the sacculus, utriculus and the semicircular canals. The semicircular canals (arc height, length, and width) tend to covary as an integrated module, whereas the utriculus and sacculus vary independently of each other. The sacculus is better integrated with the semicircular canals than the utriculus (*Figure 11*), suggesting a modular dissociation.

Taken together, our analyses suggest that the olfactory and labyrinth regions are more plastic and may reflect species differences in chemosensory and vestibular sensitivity (and potentially auditory) abilities, whereas the hindbrain remains more conserved. These findings are of note as the olfactory region seems to be changing independently from all other endocast variables, despite elongated dermal crania being modified in different ways (e.g. elongation of the dermal snout, cheek, or jaws). A similar finding has been established for extant cartilaginous fishes, where the olfactory bulbs maintain a substantial level of allometric independence from the rest of the brain in more than 100 species, reflecting differences in ecological niche, particularly habitat (*Yopak et al., 2015*; *Yopak et al., 2019*), a relationship which is common to other vertebrates (*Yopak et al., 2010*). This suggests that these species of lungfish may have had different olfactory abilities for various survival tasks including localizing prey, avoiding predators, and chemosensory communication with conspecifics.

The inner ears are comprised of a superior division containing the semicircular canals, which are predominantly involved in balance perception, and whose morphology is considered closely correlated with an animal's orientation within the water column, sensitivity to movement and even visual acuity (*Kemp and Christopher Kirk, 2014*; *Malinzak et al., 2012*; *Spoor et al., 2002*; *Spoor et al., 2007*), although the generality of this relationship is being challenged in some more recent works (*Bronzati et al., 2021*). In contrast, the inferior division, or vestibule, contains the sacculus and utriculus, which are sensitive to mechanical acceleration, but also play an auditory role in discriminating and localising sound. Inner ear morphology in fishes tends to be relatively conserved, and when there is variation it is more commonly seen in this inferior division rather than the semicircular canals (*Ladich and Schulz-Mirbach, 2016*). The sacculus (combined with the lagena in lungfish), is most sensitive to movements in the vertical plane, whereas the utriculus detects angular and linear acceleration predominantly in the horizontal plane (*Platt et al., 2004*). Similar bioimaging techniques and consequent morphometric analyses have been used to successfully predict both auditory and vestibular abilities in other aquatic vertebrates (*Lauridsen et al., 2011*; *Ramcharitar et al., 2004*), thus lending support to our functional interpretations in lungfish contained herein.

With soft tissues, such as brains, typically not fossilising, there is a need to use endocasts as proxies to infer or reconstruct gross neural morphology in extinct taxa (*Clement et al., 2016b*). With respect

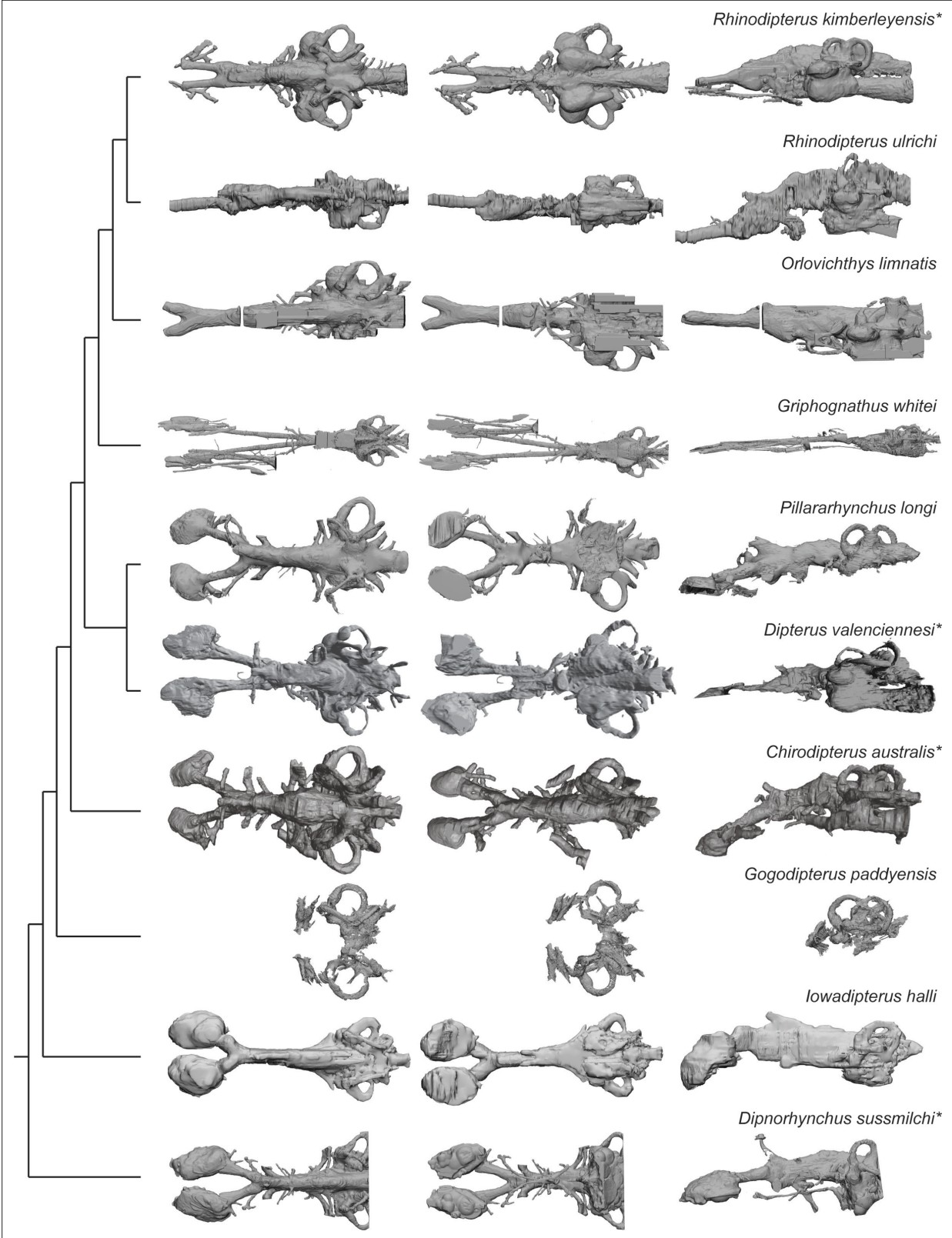

**Figure 12.** Palaeozoic lungfish endocasts in dorsal, ventral, and left lateral views, placed onto phylogeny adapted from **Challands et al., 2019** . *The following endocasts were first published elsewhere but new images were created for inclusion in this figure: *Rhinodipterus kimberleyensis* (**Clement and Ahlberg, 2014**) **Figures 2 and 3**; *Dipterus valenciennesi* (**Challands, 2015**) **Figures 6 and 9**; '*Chirodipterus*' *australis* (**Henderson and Challands, 2018**) **Figures 3–4**; *Dipnorhynchus sussmilchi* (**Clement et al., 2016a**) **Figures 2–4**.

to making meaningful biological inferences from lungfish endocasts, evidence from extant lungfish show a moderate-high correspondence between brain and endocast in life (*Challands et al., 2020*; *Clement et al., 2015*). Interestingly, there tends to be considerably higher correspondence in the forebrain and labyrinth regions (*Clement et al., 2015*), and we thus can take more confidence in potential functional interpretations relating to those regions in extinct taxa. Intriguingly, it is in those regions where we see the largest variations in our analyses, highlighting the potential to elucidate significant functional insights into an animal's sensory capabilities.

Thus, guided by our findings and the Principle of Proper Mass (*Jerison, 1973*), we believe we can draw some reasonable conclusions about brain evolution in lungfish. The hindbrain region appears to have remained relatively conserved throughout lungfish evolution, and there is no evidence of hypertrophy as is seen in some other lineages, that is within the Chondrichthyes and Actinopterygii (*Nieuwenhuys et al., 1998*; *Striedter and Northcutt, 2020*). The midbrain similarly seems to have remained relatively small and it seems unlikely that vision has ever been a dominant sense in lungfish (in contrast to many actinopterygians).

In contrast, the forebrain underwent early expansion in some of the oldest members of the lineage and appears to have continued to present day, suggesting an ongoing and increasing reliance in the role of olfaction. Relatedly, the high variability in this region throughout their evolutionary history highlights continuing modification of and importance in the role of olfaction in lungfish.

And finally, the changes seen in the labyrinth region, particularly the expansion of the utriculus, suggests increasing sensitivity to movements in the horizontal plane, perhaps reflecting changing sensory requirements from one niche to another. For example, we consider that animals living in a shallow, near-shore environment would have less requirement for sensitivity to large changes in the vertical plane compared to nektonic animals living in deeper, open environments throughout the water column. Thus, the modular dissociation between the sacculus +semicircular canals compared to the expansion of the utriculus may potentially be capturing a change from deeper water marine (nektonic) environments (as per some of the earliest lungfish, e.g. *Dipnorhynchus*), to gradually more near-shore/terrestrial ecosystems (a trend seen among Mid-Late Devonian members to present day taxa). However, we acknowledge that it is difficult to determine if increased relative utricular size results from greater reliance of sensitivity in the horizontal plane alone, or if it expands to compensate for example relative stagnation of the sacculus + semicircular canals in some way. Further studies, such as investigation of neuronal densities in extant lungfish labyrinths, may potentially help to clarify this uncertainty in future.

## Taxonomy of the 'chirodipterids'

The genus *Chirodipterus* was erected by *Gross, 1933*, and the first lungfish endocast to be described was that of *C. wildungensis* from the Upper Devonian in Germany (*Säve-Söderbergh, 1952*). Since that time, several more species of *Chirodipterus* have been described, including forms from Europe (*Mörs, 1991*; *Säve-Söderbergh, 1952*), China (*Song and Chang, 1991*), USA (*Schultze, 2010*) and Australia (*Kemp, 2000*; *Miles, 1977*). Other so-called 'chirodipterids' (sensu *Friedman, 2007*) include *Pillarahynchus* (*Barwick and Campbell, 1996*) and *Gogodipterus* (*Long, 1992*) from Australia, and *Sorbitorhynchus* (*Wang et al., 1993*) from China, suggesting a wide global distribution.

However, recently was been noted that *Chirodipterus* may in fact form a paraphyletic group, as first noted by *Friedman, 2007*, with the various species failing to be recovered as closely related in subsequent phylogenetic analyses (*Challands et al., 2019*; *Clement, 2012*; *Qiao and Zhu, 2009*). The use of endocast characters to infer phylogeny has proved valuable in tracing the evolution of lungfish (*Clement et al., 2016a*; *Friedman, 2007*), and so further analysis of 'chirodipterid' endocasts is likely to provide valuable data to clarify the taxonomy of this problematic genus.

The reconstruction and interpretation of the endocast of *Chirodipterus wildungensis* by *Säve-Söderbergh, 1952* was a valuable contribution to palaeoneurology as the first account of the endocranial structure of a fossil dipnoan. Yet, by the author's own admission, it was heavily influenced by comparison with the extant Australian lungfish (*Neoceratodus*). His interpretation that *C. wildungensis* had "a cranial anatomy astonishingly closely comparable to that of… *Neoceratodus*" *Säve-Söderbergh, 1952*, pg. (28), and subsequent conclusion by *Stensiö, 1963*, pg. (82) that its "cranial anatomy … agrees surprisingly well with *Neoceratodus*" are at odds with the morphology of 10 Devonian taxa analysed in this work and recent studies of extant Australian lungfish endocrania

(*Challands et al., 2020*; *Clement et al., 2015*). We argue that it seems more likely that the similarities with extant lungfish were (unintentionally) overstated owing to the lack of comparative material at the time, and the true morphology of *C. wildungensis* was probably more similar to that of other Devonian lungfish. Sadly, the condition of the original specimen precludes the possibility of us adequately testing this hypothesis at this time.

Two of the specimens included in our study, *Pillararhynchus longi* and *Gogodipterus paddyensis*, possess the snub-nosed morphology commonly found among 'chirodipterids', and four other taxa are known from 3D-analyses of preserved material, which should enable their endocasts to be reconstructed. Of these six 'chirodipterids' (also those that are commonly included in phylogenetic analyses of Palaeozoic lungfish), the endocast of '*Chirodipterus*' *australis* is already known (*Henderson and Challands, 2018*). *Pillararhynchus longi* and *Gogodipterus paddyensis* are investigated as part of this study, and *Chirodipterus wildungensis* was studied and CT scanned but unfortunately found not to be amenable to further study (Clement, pers. obs.). This leaves the two Chinese taxa, *Sorbitorhynchus deleaskitus* and *Chirodipterus liangchengi,* currently under examination (T. Qiao, IVPP, pers. comm. August 2019).

Our findings, and data from the Chinese taxa once published, should thus enable definitive clarification of the taxonomy of *Chirodipterus*. Nevertheless, from our current work we can already identify several characters common to 'chirodipterids' (such as a prominent pineal process, mediolaterally-oriented nasal capsules, a ventrally-directed hypophyseal region, near-equidimensional circular arcs of all three semicircular canals), which should be tested in any further studies of chirodipterid taxonomy.

## Elongated crania

At the other end of the morphological spectrum from the 'chirodipterids', the development of an elongated snout and/or skull has occurred convergently across many genera of actinopterygians but is not a well-established morphology in piscine sarcopterygians. In fact, lungfish are the only sarcopterygian group that developed significant cranial elongation, and those that did were mostly confined to the Middle to Late Devonian (ranging from the Givetian to the Famennian).

Of those that developed this condition, *Schultze, 1992* recognised that it has been achieved in a number of different ways: elongation of the anterior dermal bones and the mandible (*Griphognathus, Soederberghia, Rhynchodipterus* and possibly *Jarvikia* and *Fleurantia*); development of a longer series of posterior dermal and cheek bones (*Iowadipterus*); or elongation of the mandibular symphysis (*Rhinodipterus*). Additionally, *Orlovichthys* possesses elongate anterior rostral bones but a short mandible and large mandibular symphysis. We collectively refer to these taxa as having elongate crania rather than being 'long-snouted' or 'long-headed' in recognition that the condition is achieved in different ways.

*Stensiö, 1963* hypothesis that the dipnoan brain type was established by the Early Devonian is somewhat surprising given that he would have been familiar with forms with elongated crania such as *Soederberghia, Rhinodipterus*, *Rhynchodipterus* and *Oervigia*. However, the renaissance in dipnoan research during the 1970s and 1980s – largely driven by Roger Miles, Hans-Peter Schultze, Ken Campbell and Dick Barwick – had not yet occurred and as such, the full diversity of forms we now recognise were unknown to him. Stensiö's assertion that "the Dipnoan brain had … remained practically unchanged" (*Stensiö, 1963*, pg. 82) was challenged recently by *Clement and Ahlberg, 2014* and is similarly tested by this study.

Devonian lungfish with elongate crania fail to form a distinct clade, but rather a paraphyletic group that have converged towards head elongation (*Figure 12*). *Schultze, 1992* recognised two groups of long-snouted dipnoans: the rhynchodipterids (*Griphognathus, Rhynchodipterus, Soederberghia*) plus *Fleurantia* and *Jarvkia* and the genus *Rhinodipterus*. These two groups are differentiated by the former being denticulated forms and the latter possessing tooth plates representing different means of feeding. Despite the elongation of the lower jaw in *Rhinodipterus*, the mechanical advantage is similar to other tooth-plated lungfish and different from forms with long crania and denticulate dentition (*Clement, 2012*).

The rhynchodipterids were resolved by *Marshall, 1986* following analysis of *Holodipterus* and *Fleurantia* which in turn were found to be sister taxa to *Jarvikia*, a grade forming the most derived Devonian dipnoans below Carboniferous forms. *Friedman, 2007* resolved rhynchodipterids as the sister group to *Chirodipterus wildungensis* in a detailed analysis that incorporated new characters

of the endocranium. More recent analyses *Challands et al., 2019* have provided a more nuanced picture by including more taxa with elongate crania and have confirmed that the rhynchodipterids form a monophyletic group (albeit placed in a more basal position), and also that the 'rhinodipterids' are monophyletic (although with the likely exception of *'R.' stolbovi*). *Fleurantia* and *Apatorhynchus* are grouped together in a relatively derived position close to the majority of Carboniferous lungfish.

Other forms with elongate crania are placed in isolated positions alongside short-snouted taxa, and interestingly *Orlovichthys* and *Jarvikia* were found to resolve in the same clade along with taxa with short crania and tooth-plated dentition. *Iowadipterus* is the most basal of the lungfish with an elongated head, representing a unique solution to the morphology.

The elongation of lungfish crania has been proposed to be related to either feeding functional morphology (*Campbell and Barwick, 1986*; *Sharp and Clack, 2013*) or air-gulping ability (*Gess and Clement, 2019* other references therein). As mentioned earlier, this cranial elongation is achieved via various approaches, either by modification of the dermal bones in the snout, cheek, or in the jaws. Thus, we expected that the neurocranium and endocasts of Devonian lungfish would have also accommodated elongation via different approaches. In contrast, we found that it is nearly always the olfactory region that is elongated regardless of which portion of the dermal skull appears to lengthen, and that the hindbrain remains relatively conserved in form. The most glaring exception to this is that of *Iowadipterus* which appears to have very short olfactory canals but a long and narrow midbrain. We believe further investigation and consideration of this unusual taxon in particular is warranted.

## Conclusions

The cranial endocasts of six Palaeozoic dipnoans (*Iowadipterus halli, Gogodipterus paddyensis, Pillararhynchus longi, Griphognathus whitei, Orlovichthys limnatis,* and *Rhinodipterus ulrichi*) are created and described from synchrotron and computed tomography (CT) data. These represent a significant addition to the other known Palaeozoic lungfish endocasts (previously only four virtual endocasts had been described from tomographic data, and four others with partial endocasts were described and reconstructed directly from observation of specimens). Morphometric analyses of 10 Palaeozoic and two extant lungfish genera are conducted using standard Principal Component Analysis (PCA), Bayesian Principle Component Analysis (BPCA) and PCA for incomplete data (InDaPCA), which show the olfactory and labyrinth regions exhibit the largest amount of variation. Functional interpretation suggests that olfaction remains one of their most dominant senses throughout lungfish evolutionary history. The phylogenetic implications for the 'chirodipterids' and long-headed lungfish are discussed, and we support incorporation of additional endocranial characters in future analyses of dipnoan interrelationships.

## Materials and methods
### Specimens examined

Six species of Devonian lungfish were examined in this study: *Gogodipterus paddyensis* (WAM 70.4.250); *Griphognathus whitei* (NHMUK PV P56054); *Iowadipterus halli* (FMNH PF 12323); *Orlovichthys limnatis* (PIN 3725/110); *Pillararhynchus longi* (ANU 49196); *Rhinodipterus ulrichi* (NRM P6609a).

### Imaging and segmentation
#### *Gogodipterus paddyensis* (WAM 70.4.250)

*Gogodipterus paddyensis* was scanned using X-ray micro-computed tomography (X-ray μCT) at the Centre for Microscopy, Characterisation and Analysis (CMCA) at The University of Western Australia (UWA) in Perth, Australia in 2019. Parameters used were 80kV and 7 W using a Versa 520 XRM (Zeiss, Pleasanton, CA, USA). A LE2 beam filter was used to mitigate beam hardening and increase contrast. Source-sample and sample-detector distances were set to –73 and 48 mm, respectively, which together with the 0.4 X objective and 2 x camera binning, resulted in a final pixel resolution of 41.52 μm. Suitable image intensity was achieved with an exposure of 1 s and a total of 2401 projections were collected through 360° for the tomography. Owing to its large size, the sample was scanned in wide stitch mode, with five vertical segments needed to cover the full height. In addition, auto referencing was disabled, with a single reference collection captured prior to scanning. Reconstruction and

stitching were achieved automatically by the software using default settings. Total scan time was 21 hr 38 min. These data were manually segmented and rendered in MIMICS v.19 (Materialise).

### Griphognathus whitei (NHMUK PV P56054)

*Griphognathus whitei* was scanned at the Natural History Museum, UK, using a Nikon Metrology HMX ST 225 micro-CT scanner. The specimen was scanned with the following parameters: Energy 210 keV, voxel size 102 microns, 180 degrees, 3142 proj, angle step 0.1 deg/proj, exposure time 0.5 sec/proj, object to detector 1170 mm. (filters: 2.5 mm Cu). The resulting image stack was rendered and segmented manually using Drishti and Drishti Paint 2.6.1 to produce a virtual 3D endocast.

### Iowadipterus halli (FMNH PF 12323)

The holotype and only known specimen (FMNH PF 12323) was scanned at The University of Texas High-Resolution X-ray CT Facility, Austin, in 2007 using the following parameters: 419 kV, 1.8 mA, 130% offset, 1 brass filter, air wedge, integration time 128ms, slice thickness 0.25 mm, S.O.D. 700 mm, 2000 views, 1 ray averaged per view, 2 samples per view, inter-slice spacing 0.25 mm, field of reconstruction 180 mm (maximum field of view 184.9855 mm), reconstruction offset 4000, reconstruction scale 4000. Total slices 192, and resultant voxel size is 0.361 mm. Segmentation was performed using MIMICS v.17 & v.19 (Materialise).

### Orlovichthys limnatis (3725/110)

*Orlovichthys limnatis* was scanned at the Palaeontological Institute, Moscow, in three sections using a Skyscan 1,172 micro-CT scanner. Each section was scanned with the following parameters: Energy 100 keV, voxel size 34.1 microns, 180 degrees, 514 projections, angle step 0.7 deg/projection, exposure time 0.79 sec/projection, object to detector 344 mm. (filters: 1 mm Al). The resulting image stacks of all specimens were combined using ImageJ and the subsequent image stack was rendered and segmented manually using Drishti and Drishti Paint 2.6.1 to produce a virtual 3D endocast.

### Pillararhynchus longi (ANU 49196)

*Pillararhynchus longi* was scanned at the Australian Synchrotron (Australian Nuclear Science and Technology Organisation, ANSTO), during experiment number AM10403 in 2016 with the following parameters: Energy 40 keV, voxel size 12.2 microns, 181 degrees, 1810 projections, angle step 0.1 degree/projection, exposure time 0.22 sec/projection, object to detector 35 mm (filters: 0.45 mm carbone +5 mm high density Carbone +10 mm high density Carbone +1 mm Al +1 mm Al). The resulting image stacks of all specimens were rendered and segmented manually using MIMICS v.18 and v.19 (Materialise) to produce a virtual 3D endocast.

### Rhinodipterus ulrichi (NRM P6609a)

*Rhinodipterus ulrichi* was scanned at the European Synchrotron Radiation Facility (ESRF) on beamline ID19 using propagation phase-contrast X-ray synchrotron radiation micro-computed tomography (PPC-SRµCT) and the attenuation protocol of *Sanchez et al., 2013* with 4 meters between the sample and detector. It was imaged with 90 keV, with 4,000 projections over 360 degrees of 0.1 s each, in half-acquisition mode. Two columns of 30 scans each were required to capture the entire specimen, which had a resultant voxel size of 30 µm. The endocast was rendered and manually segmented in Drishti Paint 2.6.1.

Note: The first lungfish to have had its endocast described (*Säve-Söderbergh, 1952*) was *Chirodipterus wildungensis*, *Gross, 1933*, which is currently housed in the Museum für Naturkunde in Berlin, Germany (MB.f 12875, holotype). One of us (AMC) visited Berlin to examine and scan this specimen in December 2018. The original 'shatter method' dissection of the specimen resulted in some ~70 separate pieces of the skull, but only about 10 sizeable pieces remain in the collection today, possibly due to loss or degradation. The largest three blocks were scanned at the µCT Lab of Museum für Naturkunde using the following parameters: "c" 140kV, 500 µA, 2000 projections, 360 degrees rotation, 0.139 pixel size; "cd" 100kV, 135 µA, 2000 projections, 360 degrees rotation, 0.139 pixel size; "f" 100kV, 135 µA, 2000 projections, 360 degrees rotation, 0.139 pixel size. Despite this, we unfortunately did not find it possible to reconstruct endocranial morphology from the scanned or remaining segments of this specimen.

## Measurements and analysis

Endocast dimensions (*Figure 7*) were taken using IC Measure on-screen calibration and measurement software (https://www.theimagingsource.com). The complete original matrix (*Source data 1*) combined 17 endocast variables examined in 14 specimens representing 12 species ('*Chirodipterus*' *australis* and *Dipterus valenciennesi* were coded for two specimens each). This included 10 Palaeozoic as well as two extant species (*Neoceratodus forsteri* and *Protopterus aethiopicus*). Both specimens of '*C*'. *australis* were included in every analysis, whereas only specimen NHMUK PV P17410 of *D. valenciennesi* was retained in our analyses (the incomplete specimen NMS G.2004-10-1 of *D. valenciennesi* was excluded because it had undergone considerable compression). Specimen NHMUK PV P56038 of '*C*'. *australis* and specimen NHMUK PV P17410 of *D. valenciennesi* are the only two specimens for which all variables are available. *Gogodipterus paddyensis* is the most incomplete specimen missing 11 measurements out of a total of 17 variables. The complete original matrix has 24.7% of missing data, while the complete matrix (excluding one specimen of *Dipterus*) has 21.2% of missing data. Data were log$_{10}$-transformed to minimize differences among taxa.

The 17 measured endocast variables are defined as follows: ANG_OLF, angle of bifurcation between olfactory canals; D_NII, depth of cranial endocast at level of n.II; D_NV, depth of cranial endocast at level of n.V; DEP_HYP_F, depth of cranial endocast at level of hypophyseal fossa; HF_NII, distance from hypophyseal fossa to n.II; NAS_CAP_L, length of nasal capsules; NII_ OLF_BIF, distance from n.II to point of bifurcation between olfactory canals; NV_HF, distance from n.V to hypophyseal fossa; NX_NV, distance from n.V to n.X; OLF_CAN_L, length of olfactory canals from nasal capsules to point of bifurcation; SAC_H, height of sacculus; SAC_L, length of sacculus; SCC_H, height of semicircular canals; SCC_L, length of semicircular canals from anterior to posterior; SCC_W, width of semicircular canals from midline to lateral canal; UTR_H, height of utriculus; UTR_L, length of utriculus.

With respect to the first morphometric variable measured (ANG_OLF), we accept that olfactory *nerves* (n.I) lie between the olfactory epithelium within the nasal capsules and the olfactory bulbs, whereas olfactory *tracts* extend from the olfactory bulbs to the telencephalon. For simplicity, we herein refer to the entire length from the anterior of the telencephalon to the nasal capsules as the olfactory *canals*. Expansions within the olfactory canals are sometimes tentatively identified as the likely position of the olfactory bulbs, and can thus inform our interpretation of different taxa as having either a sessile or pedunculate arrangement. However, the angle of bifurcation between the olfactory canals is independent from olfactory bulb position in our analyses.

We used ordination methods to describe the variation and relationships among the endocranial variables. Preliminary standard principal component analyses (PCA) were performed on the correlation matrix (due to the different units of measurement used) of log$_{10}$-transformed data to maximize either the number of taxa (PCA-taxa; 9 taxa including both specimens of *C. australis*) or the number of variables (PCA-characters; 13 variables). Results from the preliminary standard PCA (PCA-taxa and PCA-characters) are briefly presented herein, but the main variation trends were congruent with both Bayesian PCA (BPCA) and PCA with incomplete data (InDaPCA). Next, in order to contend with the incomplete data, we employed two methods: one with imputation (using assignment of values by inference), that is the Bayesian principal component analysis (BPCA), and one without imputation, that is PCA for incomplete data (InDaPCA). Both methods are designed to accommodate upwards of 15% of missing data. The broken-stick method was used to determine the number of non-trivial axes to be interpreted; broken-stick values are provided with the loadings of each analysis (*Source data 2*, *Source data 3*, *Source data 4*, *Source data 5*).

Missing data were estimated using the Bayesian method developed by *Oba et al., 2003* and implemented in the R package 'pcaMethods' (*Stacklies et al., 2007*). A cross-validation method was used to estimate the number of meaningful components – in this case the two first components (*Stacklies et al., 2007*). The InDaPCA method used was designed by *Podani et al., 2021* and consists of a modified version of an eigenanalysis-based PCA for data containing missing values. The InDaPCA was performed in R using the correlation matrix. In this case, the four first components were considered for interpretation. The package 'ggplot2' in R was used to create the graphics.

## Data availablility

All data generated or analysed during this study are included in the manuscript, supporting files (5 figures and 5 tables), or via a relevant online repository. Scan data from *Chirodipterus wildungensis*,

*Gogodipterus paddyensis, Iowadipterus halli, Pillararhynchus longi* and *Rhinodipterus ulrichi* can be found on MorphoSource at: https://www.morphosource.org/projects/000381944?locale=en, *Griphognathus whitei* and *Orlovichthys limnatis* have been deposited on Dryad: https://doi.org/10.5061/dryad.2rbnzs7p8.

## Acknowledgements

We thank Kristin Mahlow (Museum für Naturkunde, Berlin, Germany), Jeremy Shaw (CMCA at University of Western Australia, Perth, Australia), Anton Maksimenko (Australian Synchrotron, Melbourne, Australia), Paul Tafforeau (European Synchrotron Radiation Facility, Grenoble, France), Matthew Colbert (University of Texas High-Resolution X-ray CT Facility), the Palaeontological Institute, Moscow, Russia, and the Natural History Museum, London, UK, for scanning specimens. Additionally, we thank Sophie Sanchez, Daniel Snitting and Vincent Dupret (Uppsala University), and Matt Friedman (University of Michigan), for scanning and software support, and/or for providing access to beamtime. We are grateful to those who hosted collection visits and enabled access to specimens, including Alexey V Pakhnevich (Russian Academy of Sciences, Moscow, Russia), Florian Witzmann (Museum für Naturkunde, Berlin, Germany), Mikael Siversson and Kate Trinajstic (Western Australian Museum, Perth, Australia), Lynne Bean and Gavin Young (Australian National University, Canberra, Australia), Zhu Min and Qiao Tuo (Institute of Vertebrate Paleontology and Paleoanthropology, Beijing, China). Finally, many thanks to our reviewers and editorial team for their detailed and positive comments about an earlier draft of this article.

This work was supported by Australian Research Council grants DP160102460 (JAL, AMC), and DP200103398 (JAL, RC, AMC, SPC), Flinders University (Impact Seed Funding to AMC, Visiting International Research Fellowships to RC & TJC), a Wallenberg Scholarship from the Knut and Alice Wallenberg Foundation (PEA, AMC), a Discovery Grant from Natural Sciences and Engineering Research Council of Canada (RC), and Callidus Services Ltd UK (TJC).

## Additional information

### Funding

| Funder | Grant reference number | Author |
|---|---|---|
| Australian Research Council | DP160102460 | Alice M Clement<br>John A Long |
| Australian Research Council | DP200103398 | Alice M Clement<br>Richard Cloutier<br>Shaun P Collin<br>John A Long |
| Flinders University | Impact Seed Funding | Alice M Clement |
| Flinders University | Visiting International Research Fellowship | Tom J Challands<br>Richard Cloutier |
| Knut och Alice Wallenbergs Stiftelse | | Per E Ahlberg |
| Natural Sciences and Engineering Research Council of Canada | | Richard Cloutier |
| Callidus Services Ltd UK | | Tom J Challands |

The funders had no role in study design, data collection and interpretation, or the decision to submit the work for publication.

### Author contributions

Alice M Clement, Conceptualization, Data curation, Formal analysis, Funding acquisition, Investigation, Methodology, Visualization, Writing - original draft, Writing - review and editing; Tom J Challands, Conceptualization, Data curation, Investigation, Methodology, Visualization, Writing - original

draft, Writing - review and editing; Richard Cloutier, Formal analysis, Methodology, Software, Writing - original draft, Writing - review and editing; Laurent Houle, Formal analysis, Visualization; Per E Ahlberg, Conceptualization, Funding acquisition, Software, Writing - review and editing; Shaun P Collin, Funding acquisition, Writing - review and editing; John A Long, Funding acquisition, Software, Writing - review and editing

### Author ORCIDs
Alice M Clement ⓘD http://orcid.org/0000-0003-0380-7347
Per E Ahlberg ⓘD http://orcid.org/0000-0001-9054-2900
John A Long ⓘD http://orcid.org/0000-0001-8012-0114

### Decision letter and Author response
Decision letter https://doi.org/10.7554/eLife.73461.sa1
Author response https://doi.org/10.7554/eLife.73461.sa2

---

## Additional files

### Supplementary files
• Transparent reporting form

• Source data 1. Table containing endocast measurements as a matrix, including additional comments and variables.

• Source data 2. Table containing PCA loadings for the taxa-optimised analyses.

• Source data 3. Table containing PCA loadings for the character-optimised analyses.

• Source data 4. Table containing loadings for the BPCA.

• Source data 5. Table containing loadings for the InDaPCA.

### Data availability
All data generated or analysed during this study are included in the manuscript, supporting files (5 figures and 5 spreadsheets), or online as follows: Scan data from Chirodipterus wildungensis, Gogodipterus paddyensis, Iowadipterus halli, Pillararhynchus longi and Rhinodipterus ulrichi can be found on MorphoSource at: https://www.morphosource.org/projects/000381944?locale=en. Griphognathus whitei and Orlovichthys limnatis have been deposited on Dryad: https://doi.org/10.5061/dryad.2rbnzs7p8.

The following datasets were generated:

| Author(s) | Year | Dataset title | Dataset URL | Database and Identifier |
|---|---|---|---|---|
| Clement AM | 2022 | Data from: Morphometric Analysis of Lungfish Endocasts Elucidates Early Dipnoan Palaeoneurological Evolution | https://doi.org/10.5061/dryad.2rbnzs7p8 | Dryad Digital Repository, 10.5061/dryad.2rbnzs7p8 |
| Clement AM | 2021 | Morphometric Analysis of Lungfish Endocasts Elucidates Early Dipnoan Palaeoneurological Evolution | https://www.morphosource.org/projects/000381944?locale=en | Morphosource, 000381944 |

---

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
