## [Editor Report]

Clement et al. described and illustrated the endocasts of six Paleozoic lungfish genera from superb 3D fossil material, which are very informative for the understanding of brain evolution of lungfishes, the extant sister group to land vertebrates. Rendering important anatomical details regarding brain evolution in lungfishes and conducting a morphometric analysis, this work will be of interest to a broad evolutionary and paleontological audience.

---

## [Decision Letter]

**Decision letter after peer review:**

Thank you for submitting your article "Morphometric Analysis of Lungfish Endocasts Elucidates Early Dipnoan Palaeoneurological Evolution" for consideration by *eLife*. Your article has been reviewed by 3 peer reviewers, including Min Zhu as the Reviewing Editor and Reviewer #1, and the evaluation has been overseen by George Perry as the Senior Editor. The following individuals involved in review of your submission have agreed to reveal their identity: Serjoscha Evers (Reviewer #2); Rodrigo Figueroa (Reviewer #3).

Essential revisions:

1) Re-organization of figures to improve the readability of the manuscript.

2) Presentation of PCA results needs further clarification.

*Reviewer #1 (Recommendations for the authors):*

1) Re-organization of figures: It will be proper to move SI-Figures 1-6 to the text. Figure 2b, whose labelings are too small to be read, can be moved to SI. Regarding Figure 3, the bottom two sub-figures can be moved to SI.

2) The authors have employed three methods to analyse the neurocranial variables. The results from BPCA and InDaPCA are overall consistent, however, there exist some deviations between preliminary classical PCAs and BPCA or InDaPCA. These differences might highlight the advantage of the new methods and need additional comments.

*Reviewer #2 (Recommendations for the authors):*

These additional comments are directed at specific paragraphs:

Line 394 : You state that variation in the labyrinth may reflect differences in auditory ability. However, the aspects of labyrinth morphology you evaluate (semicircular canals, utricle, sacculus) are primarily involved in the vestibular function of the inner ear, and not so much in hearing. In fact, in later parts of the discussion you interpret utricular variation as changing the sensitivity for movements of along the horizontal plane – which is a vestibular function, not an auditory one. Thus, I suggest you replace "auditory ability" with "vestibular sensitivity".

Paragraph line 404: Here you cite key papers regarding the ecomorphological links of labyrinth morphology (including size) and ecology. I suggest you add the paper of Spoor et al. (2007, PNAS) to this list, which otherwise is a good choice of papers to cite here. In addition, I would like to add that the "agility hypothesis" of labyrinth morphology has been questioned recently with data from non-mammalian study systems, particularly birds (Benson et al. 2017, J. Anat.) and archosaurs (Bronzati et al. 2021; Curr. Biol.). So, I would argue that vestibular form-function relationships in 2021 are actually less well understood that we thought back in say 2015. However, this also offers a good chance for you regarding this manuscript: in order to better understand vestibular ecomorphology, a broader taxonomic approach needs to be implemented, and the data you provide here is very welcome in this regard. In terms of the present text, you may want to think about modifying the paragraph in a way that say that vestibular size and shape is often interpreted as functionally tightly integrated with aspects of locomotion (such as agility), but that more recent data questions the generality of these hypotheses across vertebrates, and that your data can help elucidating these relationships further.

Line 415: The citation of Willis et al. 2013 is in my opinion not appropriate here, because these authors studied the tympanic middle ear cavities of turtles with regard to the ecological evolution of turtles, but in your sentence, you state that "similar bioimaging techniques and consequent morphometric analyses have been used to successfully predict both auditory and vestibular abilities in other aquatic vertebrates". Besides, much of the results of the Willis et al. (2013) paper has been revised and critically re-addressed by Foth et al. (2019, J. Anat.). I think the easiest solution is to ditch the citation.

Paragraph 417: In this paragraph, you state that there is a "moderate correspondence between brain and endocast in life", citing the cool Clement et al. (2015) study. As I usually work on a different group of animals, I was actually surprised how detailed the correspondence seems to be – if you compare the Clement et al. (2015) results with the brain-endocast comparisons of a turtle (Evers et al. 2019, Zool. J. Linn. Soc.) or a croc (Lessner and Holliday 2020, Anat. Rec.) you will see that the correspondence is quite high, similar to snakes (e.g. Macri et al. 2019, Nat. Comms.) or macrocephalic tetrapods like mammals or birds.

Line 435ff: It is not immediately clear, why increased utricular sizes should "reflect changing sensory requirements from deeper water marine environments […] to more terrestrial ecosystems". First, as the utricle detects linear accelerations in the horizontal plane, it is not clear why water depth should play a role (wouldn't we expect saccular changes for this, as the saccule detects linear accelerations in the vertical plane?). Also, as the change from sea to the terrestrial realm essentially is a transition from a 3D environment onto a 2D plane, wouldn't we expect changes to be concentrated in the semicircular canals themselves, as they detect angular accelerations?

Figures 2 and 3 currently lack panel labels (A, B, C,.…), but you might want to add them for clarity. The respective figure captions also suggest they should be there.

*Reviewer #3 (Recommendations for the authors):*

The following section is organized following the manuscript sections and text order. The paper is well written and organized, as well as presenting relevant and novel work. Therefore, please consider the following as simple suggestions for improving the manuscript, but not necessary for the work to be accepted for publication.

Introduction:

Line 39-41: Could probably cite papers from Jarvik, Orgiv and Stensio who provided detailed descriptions of the neurocranial morphology of several fossil fishes and are easily within the field of Paleoneurology. Since this work deals with fossil fishes, it seems important to have these represented in the introduction.

Line 42: Also crocodiles, see Watanabe et al. (2018)

Line 45: I feel that somewhere in this introduction, maybe at the end of the first paragraph, there should be some brief mention of the work from the first author that indicates a relatively high 'fit' between endocast and brain sizes in an extant lungfish. This would help making the argument of relevance of studying the lungfish endocast in the fossil record as more than simply taxonomic work.

Line 59: Probably mention here that despite the problematic homology of dermal bones, the endocast anatomy is widely comparable among vertebrates and therefore a good source for comparative analysis.

Lines 61-67: Probably double-check the zoological nomenclature for citing these taxa. Citing the authors between parenthesis after the scientific names gives the impression that these have been revised/modified.

Line 81-82: The pedunculate olfactory bulbs have probably independent acquisitions in most if not all of the clades mentioned. Additionally, note that several crown actinopterygians (e.g. some cyprinids) have pedunculate olfactory bulbs as well.

Description:

I don't have many comments about this section as it is clear and well-written. I would only recommend that structures that are described in the text should be indicated in the corresponding supplementary figures. Many of the important structures mentioned here are not labeled in the figures, making it hard for non-specialists to follow these anatomical descriptions.

Line 150: Is there any clear boundary between the forebrain and midbrain of Iowadipterus? It is not clear based on the supplementary figures provided.

Line 160: Additionally, both anterior and posterior semicircular canals seem to be laterally compressed, which is quite unusual.

Line 192: Describe ampullae shape and position in this and in other taxa described here.

Principal Component Analysis:

The methodology used is clearly stated and is compatible with the data available. However, the use of measurements (e.g. length of the inner ear; angle between the olfactory canals) could preclude the analysis from obtaining a better representation of the complex shapes of lungfish endocasts. Thus, a morphometric analysis using landmarking of homologous features of these endocasts could provide a broader perspective on their similarity/dissimilarity. I understand however that this would be a time-consuming effort and is beyond the scope of the exploratory analysis presented here.

Line 293: "no methods of imputation were used" meaning unclear.

Line 346-349: The clustering of meristics of the same structure are expected, but the cluster of depth of nerves II and V is interesting, especially taking into account they emerge from different regions of the endocast/brain.

Discussion:

Line 392-394: But what about ontogeny? How sure can you be that the specimens used are all of comparable age/development? There is considerable ontogenetic variation and heterocrony in the neurocranium of vertebrates, which could potentially bias some of the interpretations shown here.

Line 402: Is there anything to say about the size and shape of the olfactory capsules in relation to the plasticity in morphology of the olfactory canals described above.

Line 411-413: So it would be interesting to see how this is distributed among the Devonian taxa.

Line 424: Could the higher fit between brain and endocast in the forebrain of extant lungfishes be directly related to the general plasticity of this region? In other words, endocranial space is being modeled based on brain anatomy rather than brain being restricted by the endocranial cavity space for that region?

---

## [Author Response]

Reviewer #1 (Recommendations for the authors):1) Re-organization of figures: It will be proper to move SI-Figures 1-6 to the text. Figure 2b, whose labelings are too small to be read, can be moved to SI. Regarding Figure 3, the bottom two sub-figures can be moved to SI.

The figures have now been reorganised to include the six endocast figures as main figures, and statistical analysis results have been modified so that they are easier to read and interpret. This has now resulted in 12 main text figures, and 5 supplementary figures.

2) The authors have employed three methods to analyse the neurocranial variables. The results from BPCA and InDaPCA are overall consistent, however, there exist some deviations between preliminary classical PCAs and BPCA or InDaPCA. These differences might highlight the advantage of the new methods and need additional comments.

We have now clarified the usage of the BPCA and InDaPCA over the standard PCA in the text and rewritten this section considerably (see Principal Component Analyses section on pages 7-9); we have changed the terminology “classical PCS” for “standard PCA”.

Reviewer #2 (Recommendations for the authors):These additional comments are directed at specific paragraphs:Line 394 : You state that variation in the labyrinth may reflect differences in auditory ability. However, the aspects of labyrinth morphology you evaluate (semicircular canals, utricle, sacculus) are primarily involved in the vestibular function of the inner ear, and not so much in hearing. In fact, in later parts of the discussion you interpret utricular variation as changing the sensitivity for movements of along the horizontal plane – which is a vestibular function, not an auditory one. Thus, I suggest you replace "auditory ability" with "vestibular sensitivity".

We have edited the sentence to now read: “…and vestibular sensitivity (and potentially auditory) abilities”

Paragraph line 404: Here you cite key papers regarding the ecomorphological links of labyrinth morphology (including size) and ecology. I suggest you add the paper of Spoor et al. (2007, PNAS) to this list, which otherwise is a good choice of papers to cite here. In addition, I would like to add that the "agility hypothesis" of labyrinth morphology has been questioned recently with data from non-mammalian study systems, particularly birds (Benson et al. 2017, J. Anat.) and archosaurs (Bronzati et al. 2021; Curr. Biol.). So, I would argue that vestibular form-function relationships in 2021 are actually less well understood that we thought back in say 2015. However, this also offers a good chance for you regarding this manuscript: in order to better understand vestibular ecomorphology, a broader taxonomic approach needs to be implemented, and the data you provide here is very welcome in this regard. In terms of the present text, you may want to think about modifying the paragraph in a way that say that vestibular size and shape is often interpreted as functionally tightly integrated with aspects of locomotion (such as agility), but that more recent data questions the generality of these hypotheses across vertebrates, and that your data can help elucidating these relationships further.

We have now added Spoor et al. 2007, in addition to a qualifying statement about the consistency of this relationship in vertebrates.

Line 415: The citation of Willis et al. 2013 is in my opinion not appropriate here, because these authors studied the tympanic middle ear cavities of turtles with regard to the ecological evolution of turtles, but in your sentence, you state that "similar bioimaging techniques and consequent morphometric analyses have been used to successfully predict both auditory and vestibular abilities in other aquatic vertebrates". Besides, much of the results of the Willis et al. (2013) paper has been revised and critically re-addressed by Foth et al. (2019, J. Anat.). I think the easiest solution is to ditch the citation.

As per Reviewer #2’s suggestion, this citation has now been removed.

Paragraph 417: In this paragraph, you state that there is a "moderate correspondence between brain and endocast in life", citing the cool Clement et al. (2015) study. As I usually work on a different group of animals, I was actually surprised how detailed the correspondence seems to be – if you compare the Clement et al. (2015) results with the brain-endocast comparisons of a turtle (Evers et al. 2019, Zool. J. Linn. Soc.) or a croc (Lessner and Holliday 2020, Anat. Rec.) you will see that the correspondence is quite high, similar to snakes (e.g. Macri et al. 2019, Nat. Comms.) or macrocephalic tetrapods like mammals or birds.

We have now changed this to “moderate-high” at the request of Reviewer #2.

Line 435ff: It is not immediately clear, why increased utricular sizes should "reflect changing sensory requirements from deeper water marine environments […] to more terrestrial ecosystems". First, as the utricle detects linear accelerations in the horizontal plane, it is not clear why water depth should play a role (wouldn't we expect saccular changes for this, as the saccule detects linear accelerations in the vertical plane?). Also, as the change from sea to the terrestrial realm essentially is a transition from a 3D environment onto a 2D plane, wouldn't we expect changes to be concentrated in the semicircular canals themselves, as they detect angular accelerations?

We have clarified this section so that our reasoning about the expanded utriculus potentially reflecting a shallower water niche should be easier to follow: “And finally, the changes seen in the labyrinth region, particularly the expansion of the utriculus, suggests increasing sensitivity to movements in the horizontal plane, perhaps reflecting changing sensory requirements from one niche to another. For example, we consider that animals living in a shallow, near-shore environment would have less requirement for sensitivity to large changes in the vertical plane compared to nektonic animals living in deeper, open environments throughout the water column. Thus, the modular dissociation between the sacculus + semicircular canals compared to the expansion of the utriculus may potentially be capturing a change from deeper water marine (nektonic) environments (as per some of the earliest lungfish, e.g. *Dipnorhynchus*), to gradually more near-shore/terrestrial ecosystems (a trend seen among Mid-Late Devonian members to present day taxa).”

Figures 2 and 3 currently lack panel labels (A, B, C,.…), but you might want to add them for clarity. The respective figure captions also suggest they should be there.

Panel labels have now been added (this was an accidental omission, thanks for picking it up!)

Reviewer #3 (Recommendations for the authors):The following section is organized following the manuscript sections and text order. The paper is well written and organized, as well as presenting relevant and novel work. Therefore, please consider the following as simple suggestions for improving the manuscript, but not necessary for the work to be accepted for publication.Introduction:Line 39-41: Could probably cite papers from Jarvik, Orgiv and Stensio who provided detailed descriptions of the neurocranial morphology of several fossil fishes and are easily within the field of Paleoneurology. Since this work deals with fossil fishes, it seems important to have these represented in the introduction.

Agreed. We have now added an additional sentence and included further references to early palaeoneurological studies in fossil fishes: “Nevertheless, there were some early works specifically investigating fossil fishes which laid the foundation for further palaeoneurological research in early vertebrates (Chang 1982; Jarvik 1942; Jarvik 1972; Stensiö 1963).”

Line 42: Also crocodiles, see Watanabe et al. (2018)

We agree that the study of Watanabe et al. 2018 is a valuable addition to understanding endocranial shape in archosaurs, although that is a study of extant taxa and thus not relevant to this statement concerning *palaeo*neurology specifically.

Line 45: I feel that somewhere in this introduction, maybe at the end of the first paragraph, there should be some brief mention of the work from the first author that indicates a relatively high 'fit' between endocast and brain sizes in an extant lungfish. This would help making the argument of relevance of studying the lungfish endocast in the fossil record as more than simply taxonomic work.

Thank you for this suggestion. We have now included an additional paragraph detailing the recent work on brainendocast relationship in extant lungfish: “While aspects of the anatomy of the central nervous system in extant lungfish have been known for close to 150 years (Collin 2007; Huxley 1876; Northcutt 1986), it was only recently that the specific spatial relationship between the brain and the endocranial cavity was investigated. This was examined first in the Australian lungfish, *Neoceratodus* (Clement et al. 2015), and later in the lepidosirenid lungfish (*Lepidosiren* and *Protopterus*) and other piscine sarcopterygians (Challands et al. 2020). It was found that contrary to earlier reports of the lungfish brain filling just a fraction (10%) of its brain cavity (Jerison 1973), values in fact ranged between 40-80% (Challands et al. 2020; Clement et al. 2021; Clement et al. 2015). Moreover, these studies highlighted that the forebrain and labyrinth regions in particular had a close correspondence between brain and endocast, in comparison with the mid and hindbrain regions with a looser association.”

Line 59: Probably mention here that despite the problematic homology of dermal bones, the endocast anatomy is widely comparable among vertebrates and therefore a good source for comparative analysis.

We have now added the following statement: “Despite the problematic homology of dermal bones, endocranial anatomy tends to be far more conserved in vertebrates and thus highly valuable for comparative analysis.”

Lines 61-67: Probably double-check the zoological nomenclature for citing these taxa. Citing the authors between parenthesis after the scientific names gives the impression that these have been revised/modified.

The cited authors are in reference to the papers referring to neurocranial information on these taxa.

Line 81-82: The pedunculate olfactory bulbs have probably independent acquisitions in most if not all of the clades mentioned. Additionally, note that several crown actinopterygians (e.g. some cyprinids) have pedunculate olfactory bulbs as well.

“Probably” is exactly why we conduct studies such as this to determine the polarity of endocast characters.

Description:I don't have many comments about this section as it is clear and well-written. I would only recommend that structures that are described in the text should be indicated in the corresponding supplementary figures. Many of the important structures mentioned here are not labeled in the figures, making it hard for non-specialists to follow these anatomical descriptions.

The figures have been updated so as to enable greater ease of interpretation by non-specialists.

Line 150: Is there any clear boundary between the forebrain and midbrain of Iowadipterus? It is not clear based on the supplementary figures provided.

The boundary between the forebrain and midbrain is considered according to the position of the cranial nerves (specifically, the midbrain region begins posterior to n.II)

Line 160: Additionally, both anterior and posterior semicircular canals seem to be laterally compressed, which is quite unusual.

The *Iowadipterus* data provided a significant challenge to segment and we do accept that a newer scan could potentially elucidate it’s anatomy more clearly than what we have been able to segment out of the current dataset. Additionally this specimen may have undergone some slight compression during preservation which could damage delicate structures such as the semicircular canals more so than other more robust regions.

Line 192: Describe ampullae shape and position in this and in other taxa described here.

Ampullae were already described adequately in the text for *Griphognathus*, and the poor scan quality preclude us from knowing the condition in *Iowadipterus*. We have added “elongate” to the description of *Orlovichthys* and *Pillararhynchus*, and the ampulla is now described as “somewhat triangular in shape” for *Rhinodipterus*.

Principal Component Analysis:The methodology used is clearly stated and is compatible with the data available. However, the use of measurements (e.g. length of the inner ear; angle between the olfactory canals) could preclude the analysis from obtaining a better representation of the complex shapes of lungfish endocasts. Thus, a morphometric analysis using landmarking of homologous features of these endocasts could provide a broader perspective on their similarity/dissimilarity. I understand however that this would be a time-consuming effort and is beyond the scope of the exploratory analysis presented here.

We agree with Reviewer #3 that 3D geometric morphometrics could have provided more sophisticated analytical methods. We have already responded to a similar comment from Reviewer #2. Namely, geometric morphometrics has some limitations with regard to the type of data that we analysed: (1) low sample size and (2) missing/incomplete data. In order to have a comprehensive coverage of the brain shape, it would have required us to have numerous landmarks (and semi-landmarks) to represent the complexity of brain shape. First, as already mentioned by Reviewer #2, our sample size (12 taxa) is low (although it is an impressive sample size when considering the type of data). Although there are no universal rules concerning the ratio “number of specimens / number of landmarks” (Zelditch et al., 2012), ideally the sample size must be from two to three times the number of landmarks. Thus, with a sample size of 12 we could have used ca. 4-6 landmarks which is very limited to describe complex shapes. In addition, in order to use geometric morphometrics (2D or 3D), the landmarks should be present on all the specimens. Because of the partial completeness of the studied fossils, the brain endocasts are not uniformly known for each species. Incomplete and deformed specimens prompt the removal of potential landmarks for analyses. Even using right-left reflexion of the endocasts, most specimens do not share all neurocranial information.

Line 293: "no methods of imputation were used" meaning unclear.

We rephrased the sentence in order to clarify the meaning of imputation

Line 346-349: The clustering of meristics of the same structure are expected, but the cluster of depth of nerves II and V is interesting, especially taking into account they emerge from different regions of the endocast/brain.

We just want to clarify that we did not used meristic variables (i.e., a quantitative count of structure) but rather morphometric variables (a measurement on a structure). We are unsure what Reviewer #3 refers to here.

Discussion:Line 392-394: But what about ontogeny? How sure can you be that the specimens used are all of comparable age/development? There is considerable ontogenetic variation and heterocrony in the neurocranium of vertebrates, which could potentially bias some of the interpretations shown here.

As we are working with exceedingly rare specimens (all fossil lungfish endocasts known), we do not have the luxury to conduct analyses on specimens of known and comparable age/stage. We acknowledge that ontogeny has considerable influence on the neurocranium and have acknowledged this within our text: “Due to the extreme age and rarity of 3D preserved lungfish endocrania, we used all material available to us. Admittedly we cannot hope to control for many factors, including ontogeny, so our interpretations must naturally take that into consideration.”

Line 402: Is there anything to say about the size and shape of the olfactory capsules in relation to the plasticity in morphology of the olfactory canals described above.

Thanks for this suggestion. We have now expanded the text about the olfactory canals with the following: “The shape of the olfactory capsules appears highly variable in Devonian lungfish endocasts (Figure 12). These are most commonly oval-shaped, although the orientation can vary. Others are highly elongate as in *Griphognathus whitei,* to triangular in *‘Chirodipterus’ australis*. Our study simply captured length but future work on shape and orientation may prove pertinent to the overall plasticity of the olfactory region.”

Line 411-413: So it would be interesting to see how this is distributed among the Devonian taxa.

Figure 12 is a good summary of the endocasts included in our study and captures how the relative proportions of the sacculus-utriculus differ among taxa.

Line 424: Could the higher fit between brain and endocast in the forebrain of extant lungfishes be directly related to the general plasticity of this region? In other words, endocranial space is being modeled based on brain anatomy rather than brain being restricted by the endocranial cavity space for that region?

This is a fascinating suggestion and we consider it likely that the endocranial space is indeed influenced by the growing forebrain. Developmental studies on extant taxa could likely show this relationship more clearly (a study that I would love to do!)